# Are Structural Concepts Universal in Transformer Language Models? Towards Interpretable Cross-Lingual Generalization

**Ningyu Xu**[1,2], **Qi Zhang**[1], **Jingting Ye**[3], **Menghan Zhang**[2,4], **Xuanjing Huang**[1,2]

[1]School of Computer Science, Fudan University

[2]Institute of Modern Languages and Linguistics, Fudan University

[3]Department of Chinese Language and Literature, Fudan University

[4]Research Institute of Intelligent Complex Systems, Fudan University

nyxu22@m.fudan.edu.cn    {qz,yejingting,mhzhang,xjhuang}@fudan.edu.cn

## Abstract

Large language models (LLMs) have exhibited considerable cross-lingual generalization abilities, whereby they implicitly transfer knowledge across languages. However, the transfer is not equally successful for all languages, especially for low-resource ones, which poses an ongoing challenge. It is unclear whether we have reached the limits of implicit cross-lingual generalization and if explicit knowledge transfer is viable. In this paper, we investigate the potential for explicitly aligning conceptual correspondence between languages to enhance cross-lingual generalization. Using the syntactic aspect of language as a testbed, our analyses of 43 languages reveal a high degree of alignability among the spaces of structural concepts within each language for both encoder-only and decoder-only LLMs. We then propose a meta-learning-based method to learn to align conceptual spaces of different languages, which facilitates zero-shot and few-shot generalization in concept classification and also offers insights into the cross-lingual in-context learning phenomenon. Experiments on syntactic analysis tasks show that our approach achieves competitive results with state-of-the-art methods and narrows the performance gap between languages, particularly benefiting those with limited resources.

## 1 Introduction

Cross-lingual generalization entails repurposing the knowledge acquired in one language to another with little supervision, thereby mitigating the digital language divide. Despite the vast variations across languages, it is possible to identify corresponding concepts among them, which provides a basis for cross-linguistic generalization (Croft, 1991; Haspelmath, 2010, 2021). This has been instantiated by frameworks such as Universal Dependencies (UD) (de Marneffe et al., 2021), where the structural concepts including word classes (e.g., "noun" and "verb") and grammatical relations (e.g., "subject" and "object") are defined in a cross-linguistically consistent way. While large language models (LLMs) have demonstrated their capacity to induce these concepts within individual languages (Tenney et al., 2019; Liu et al., 2019; Chi et al., 2020; Linzen and Baroni, 2021), they encounter difficulties in generalizing the knowledge across languages (Joshi et al., 2020; Blasi et al., 2022; Majewska et al., 2022). This raises questions about whether LLMs are able to capture the underlying conceptual correspondence and how to harness the knowledge for improved generalization.

Previous work has shown that cross-linguistic similarities are automatically captured in the representation space of LLMs, enabling zero-shot cross-lingual transfer (Pires et al., 2019; Wu and Dredze, 2019; Chi et al., 2020; Papadimitriou et al., 2021; Muller et al., 2021; Xu et al., 2022). Efforts have been made to further enhance their generalization by exploiting high-resource languages for parameter and information sharing (Üstün et al., 2020; Nooralahzadeh et al., 2020; Choenni et al., 2023) or enforcing alignment between languages (Cao et al., 2019; Schuster et al., 2019; Sherborne and Lapata, 2022), whereas these approaches typically rely on structural and lexical similarities between languages and fall short when dealing with low-resource languages distant from high-resource ones (Ponti et al., 2021; de Lhoneux et al., 2022). With the ever-increasing size of LLMs, recent work has explored methods to elicit the multilingual ability of LLMs via in-context learning (Winata et al., 2021; Tanwar et al., 2023), alleviating the cost of parameters updates, but the generalization performance lags behind and is highly sensitive to prompt design (Lai et al., 2023; Ahuja et al., 2023). Overall, the generalization is predominantly realized implicitly and it remains unclear whether the commonalities shared across languages have been fully exploited.

In this paper, we investigate the potential to explicitly leverage the conceptual correspondence between languages for cross-lingual generalization. We focus on the structural concepts outlined in UD, which are at the core of syntactic analyses across languages (Croft, 1991) and have shown to be learned by LLMs, offering a valuable testbed for our analyses. For each language, we learn a linear transformation of an LLM's representation space, which defines a conceptual space where samples can be classified by their distances to prototypes for each concept. Concepts represented by the LLM are then regarded as clusters discernible from others based on their prototypes. Analyses across 43 typologically distinct languages reveal a high degree of alignability among their conceptual spaces, indicating that the conceptual correspondence is implicitly established in LLMs (Section 2). We then present a meta-learning-based method that learns to explicitly align different languages with limited data available, facilitating zero-shot and few-shot generalization in concept classification. We demonstrate the effectiveness of our approach for both encoder-only (Section 3) and decoder-only LLMs (Section 4), achieving encouraging results especially for low-resource languages.

In summary, our contributions are as follows: 1) We demonstrate that the conceptual correspondence between languages, in terms of the structural concepts defined in UD, is implicitly established in both encoder-only and decoder-only LLMs. 2) We propose a meta-learning-based approach to explicitly aligning conceptual correspondence between different languages, enabling cross-lingual generalization in zero-shot and few-shot scenarios without requiring parameter updates to the LLMs. Our method achieves competitive results with state-of-the-art methods and reduces the performance gap between languages, particularly benefiting low-resource ones. 3) Our approach provides insights into the cross-lingual in-context learning phenomenon. Integrated with the prompt-based learning paradigm, it achieves promising gains in generalizing to novel languages.[1]

## 2 Correspondence between Structural Concepts within Different Languages

This section investigates whether Transformer-based LLMs are able to induce the conceptual

---

[1]Our code is available at https://github.com/ningyuxu/structural_concepts_correspondence.

correspondence between different languages from plain text, which could lay the foundation for better cross-lingual generalization. Concretely, we first derive structural concepts within individual languages, and then evaluate whether these concepts are readily alignable across languages.

### 2.1 Method

**Deriving concepts**   Let $\mathcal{D} = \left\{ \mathbf{x}_i, y_i \right\}_{i=1}^{N}$ denote a dataset consisting of $N$ feature-label pairs in a language $L$, where the features $\mathbf{x}_i \in \mathbb{R}^n$ are $n$-dimensional representations yielded by LLMs and the labels $y_i \in \{1, \cdots, K\}$ are the corresponding structural concept, and $\mathcal{D}_k$ is the set of samples with label $k$. We compute an $m$-dimensional prototype $\mathbf{c}_k \in \mathbb{R}^m$ for each concept $k$ by learning a linear transformation $A \in \mathbb{R}^{n \times m}$, such that

$$\mathbf{c}_k = \frac{1}{|\mathcal{D}_k|} \sum_{(\mathbf{x}_i, y_i) \in \mathcal{D}_k} A\mathbf{x}_i, \qquad (1)$$

and the label of a feature $\mathbf{x}$ can be identified with respect to its distances to the prototypes in the transformed representation space. Specifically, the probability that $\mathbf{x}$ is an instance of concept $k$ is given by

$$p_A\left(y = k \mid \mathbf{x}\right) = \frac{\exp\left(-d\left(A\mathbf{x}, \mathbf{c}_k\right)\right)}{\sum_{k'} \exp\left(-d\left(A\mathbf{x}, \mathbf{c}_{k'}\right)\right)}, \quad (2)$$

where $d\left(\cdot, \cdot\right)$ is the squared Euclidean distance. The parameters of the transformation, the matrix $A$, is learned by minimizing the negative log probability $\mathcal{L}_A = -\log p_A\left(y = k \mid \mathbf{x}\right)$ of the gold concept $k$ through gradient descent. Our probing method is inspired by Prototypical Networks (Snell et al., 2017), but we constrain the transformation to be linear as our goal is to investigate the geometry of LLMs, akin to Hewitt and Manning (2019).

**Measuring Alignability**   We employ two complementary methods to measure the alignability between structural concepts in different languages: representational similarity analysis (RSA) (Kriegeskorte et al., 2008) and Procrustes analysis. RSA is non-parametric and has been widely used for measuring the degree of topological alignment between representation spaces based on their (dis)similarity matrices (Kriegeskorte and Diedrichsen, 2019). Procrustes analysis evaluates the extent to which two spaces can be aligned linearly by finding the optimal orthogonal transformation.

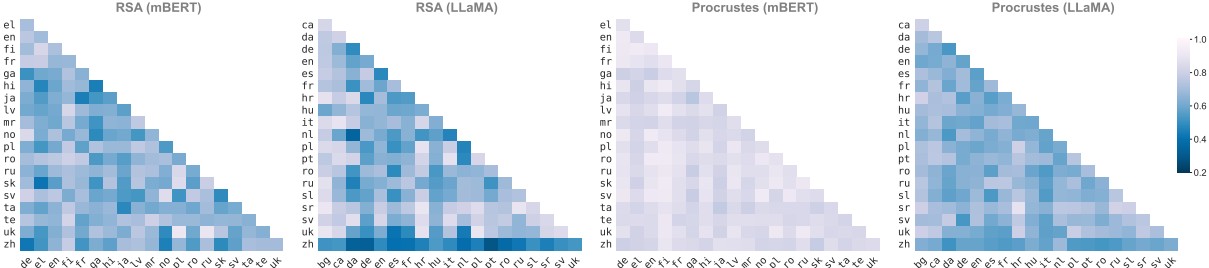

Figure 1: Alignability between structural concepts (word classes) in different languages measured by RSA and Procrustes analysis, which is significantly higher than baselines. (The results for all languages, along with the alignability between grammatical relations, are presented in Appendix A.2.)

Given two languages $L_1$ and $L_2$ with $K$ shared structural concepts, we derive prototypes for each concept, which serve as parallel points that allows for comparison among different languages. Infrequently used concepts with fewer than 20 samples are excluded for our analysis. For **RSA**, we compute a dissimilarity matrix $M \in \mathbb{R}^{K \times K}$ for each language, where the entry $M_{i,j} = d(\mathbf{c}_i, \mathbf{c}_j)$ is the distance between the $i^{\text{th}}$ and $j^{\text{th}}$ prototypes ($1 \leq i, j \leq K$). The alignability is computed as the Spearman's correlation between the lower diagonal portion of the two matrices, ranging from $-1$ to $1$. For **Procrustes analysis**, we evaluate the fitness of the linear transformation through the average proportion of explained variance.

## 2.2 Setup

**Model** We use Multilingual BERT (mBERT) (Devlin et al., 2019) and LLaMA 7B model (Touvron et al., 2023) for our experiments. Both are pretrained on multiple languages without explicit cross-lingual information, enabling us to probe the cross-linguistic knowledge induced exclusively from raw text. A linear transformation $A \in \mathbb{R}^{n \times m}$ with varying $m$ is trained to project the features $\mathbf{x} \in \mathbb{R}^n$ yielded by the LLM into an $m$-dimensional space, whereby we test what rank of transformation is needed to extract structural concepts.

**Data** The data used in all our experiments is from UD v2.10 (Zeman et al., 2022). For mBERT, we select 43 typologically distinct languages that represent a diverse range of language families. For LLaMA, we test it on 20 languages, including one that is not included in its pretraining corpus for comparison.[2]

**Baselines** The alignability between the structural concepts in two languages should contrast with

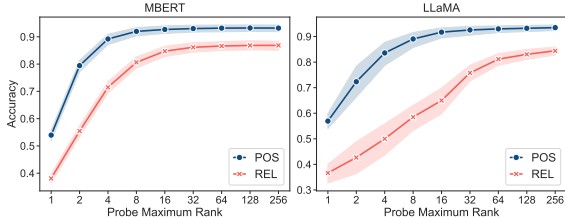

Figure 2: Accuracy in identifying word classes (POS) and grammatical relations (REL) averaged across different languages, with the linear transformation constrained to varying maximum dimensionality. The colored bands denote 95% confidence intervals. Structural concepts can be identified based on prototypes within a relatively low-dimensional space.

cases where the structure of their spaces is deformed. Given the prototypes for $K$ structural concepts derived from two languages, we construct the following baselines: (i) **RP**, which randomly swaps each prototypes for another in one of the two languages; (ii) **RC**, where we randomly select a sample of each concept instead of their prototypes in one language; (iii) **RS**, where we randomly select $K$ samples in one language.

## 2.3 Results

**Structural concepts can be identified based on prototypes** The structural concepts including word classes and grammatical relations can be successfully distinguished according to their distances to the prototypes (Figure 2)[3]. The structural information can be effectively encoded within a relatively low-dimensional space, which varies across different models. We note that more expressive probing models are needed to extract structural concepts, especially grammatical relations, from LLaMA; we leave exploration of

---

[2]See Appendix E for the languages and treebanks we use.

[3]We take the $7^{\text{th}}$ layer of mBERT here as it is most effective in encoding structural information (Appendix A.1).

this to future work.

**Structural concepts are readily alignable across languages** Figure 1 depicts the alignablity between the structural concepts in different languages. Both word classes and grammatical relations are highly correlated across languages and can be approximately aligned through an orthogonal transformation (rotation, reflection, etc.). Moreover, the alignability is significantly higher than baselines, reinforcing that the conceptual correspondence between languages is reflected in the representation space.

## 2.4 Discussion

It has been suggested that word embeddings in different languages are approximately isometric and can be aligned through a linear transformation (Mikolov et al., 2013; Lample et al., 2018; Schuster et al., 2019). However, the meaning of each individual words, rather than the underlying concepts (Youn et al., 2016; Xu et al., 2020), might not be indeed alignable across languages (Thompson et al., 2020), and enforcing such alignment can hurt downstream performance (Glavaš et al., 2019; Wu and Dredze, 2020). We here propose to establish the alignment based on conceptual correspondence that can serve as yardsticks for cross-linguistic comparison (Haspelmath, 2021), and the structural concepts defined in UD are generally designed to meet the need.

The alignability of structural concepts across different languages is relatively consistent, providing evidence that their correspondence implicitly encoded in LLMs, though not well aligned. However, variations remain between different language pairs. Besides subtle cross-linguistic differences with regard to the structural concepts (e.g., Ponti et al., 2018), this might result from i) the lack of sufficient data in the UD treebank for approximating the prototypes, and ii) the degenerate representation spaces of certain languages, which has been attributed to factors including inadequate pretraining data and deficiencies in tokenization methods for specific languages (Vulić et al., 2020; Rust et al., 2021; Blaschke et al., 2023; Purkayastha et al., 2023). The disparities among languages are also reflected in our experiment results regarding the classification of structural concepts, especially for languages not well represented in the pretraining corpora[4].

---

[4]See Appendix A.1 for details.

## 3 Aligning Conceptual Correspondence for Cross-Lingual Generalization

The previous section shows that the universal structural concepts are readily alignable in the LLMs. Next, we investigate how to leverage the knowledge for cross-lingual generalization. We rely on meta-learning to learn to align conceptual correspondence between different languages in both zero-shot and few-shot scenarios, analyzing how different factors including the number of available examples and the languages used for meta-training may impact the generalization.

### 3.1 Method

**Learning to align with a few examples** We first derive prototypes $\mathbf{c}_k^{\mathcal{S}}$ from a source language $L_{\mathcal{S}}$ following the method in Section 2.1, and then learn to align samples in different languages with $\mathbf{c}_k^{\mathcal{S}}$ via meta-learning. We employ a composite function $F = g_\alpha \circ f_\phi$ to establish the alignment. The function $f_\phi : \mathbb{R}^n \rightarrow \mathbb{R}^m$ with parameters $\phi$ is language-agnostic and projects features yielded by LMs into an $m$-dimensional space, where samples belonging to each concept in a target language $L_{\mathcal{T}}$ are expected to cluster around their prototypes $\mathbf{c}_k^{\mathcal{T}}$. As shown in Section 2.3, an orthogonal transformation suffices to align the prototypes in two languages while preserving the geometry of the original spaces. We thus use a language-specific linear mapping $g_\alpha : \mathbb{R}^m \rightarrow \mathbb{R}^m$ to convert $\mathbf{c}_k^{\mathcal{T}}$ into prototypes $\mathbf{c}_k^{\mathcal{S}}$, which allows us to identify the structural concepts according to

$$p_F\left(y = k \mid \mathbf{x}\right) = \frac{\exp\left(-d\left(F(\mathbf{x}),\mathbf{c}_k^{\mathcal{S}}\right)\right)}{\sum_{k'} \exp\left(-d\left(F(\mathbf{x}),\mathbf{c}_{k'}^{\mathcal{S}}\right)\right)}. \quad (3)$$

The parameters are together optimized by minimizing the negative log-probability of the gold concept $k$. We use labeled data in multiple languages to learn the function $F$ during meta-training. The language-agnostic function $f_\phi$ is optimized over the entire training procedure, and the language-specific function $g_\alpha$ is learned separately for different languages. During meta-testing, $f_\phi$ is kept fixed while $g_\alpha$ is learned from scratch using the provided examples.

**Aligning with unified prototypes for zero-shot generalization** In the zero-shot setting, instead of being given a few examples to learn the alignment, we rely on meta-learning to establish unified prototypes for each concept. We learn a linear

| | sr† | ga† | be | br* | cy† | fo* | gsw* | kk | mr† | pcm* | sa* | ta† | te† | tl | wbp* | yo | AVG | STD |
|---|---|---|---|---|---|---|---|---|---|---|---|---|---|---|---|---|---|---|
| UDAPTER | - | - | **96.9** | 72.2 | 69.7 | 79.6 | 65.9 | **83.4** | 66.5 | 54.7 | 42.2 | 70.3 | 84.2 | 78.4 | 34.1 | 63.7 | 58.4 | 0.214 |
| M28-0 | 92.2 | 71.4 | 90.3 | 75.4 | 69.9 | 84.6 | 64.7 | 77.2 | 79.3 | 37.7 | 45.2 | 73.1 | 78.2 | 73.8 | 51.6 | 61.0 | 56.6 | 0.212 |
| M28-10 | 94.3 | 73.9 | 89.3 | 79.4 | 73.4 | 85.1 | 68.8 | 79.0 | 80.1 | 43.8 | 49.4 | 77.5 | 80.2 | 80.5 | 54.6 | 63.1 | 59.0 | 0.208 |
| M28-50 | 95.2 | 75.9 | 90.7 | 81.1 | 76.9 | **86.3** | 70.6 | 79.8 | 79.0 | 64.3 | **52.0** | 79.5 | 80.3 | **83.7** | **58.8** | **67.0** | 62.5 | 0.183 |
| M43-0 | 96.2 | 86.8 | 90.3 | 73.9 | 88.5 | 83.8 | 63.4 | 77.7 | 87.5 | 47.3 | 46.0 | 81.3 | **90.4** | 70.0 | 49.3 | 58.0 | 57.9 | 0.228 |
| M43-10 | 95.8 | 86.6 | 89.5 | 80.3 | 87.3 | 84.9 | 69.0 | 80.2 | 86.7 | 54.5 | 48.7 | **81.9** | 90.0 | 78.6 | 53.9 | 62.2 | 60.8 | 0.211 |
| M43-50 | **96.4** | **87.0** | 91.6 | **82.3** | **88.9** | 85.7 | **72.2** | 80.3 | **88.6** | 70.6 | 49.8 | 81.5 | 90.0 | 82.8 | **58.8** | 66.7 | **64.4** | 0.187 |

Table 1: The zero-shot and few-shot generalization performance on POS tagging for a subset of languages unseen during meta-training and low-resource languages. Languages marked with "∗" are not included in the pretraining corpus. "†" indicates that the language is involved in meta-training for M43. AVG and STD denotes the average accuracy and standard deviation respectively for 30 low-resource languages. (The results for all languages, together with the performance on the classification of grammatical relations, are given in Appendix B.)

mapping $h_\omega : \mathbb{R}^m \rightarrow \mathbb{R}^m$ to convert $\mathbf{c}_k^{\mathcal{S}}$ to unified prototypes $\mathbf{c}_k^\omega = h\left(\mathbf{c}_k^{\mathcal{S}}\right)$ and use a language-agnostic function $f_\phi$ to match samples with them. The classification is then performed based on $d\left(f_\phi(\mathbf{x}), \mathbf{c}_k^\omega\right)$. We optimize the parameters during meta-training and directly apply the models to other languages for meta-testing.

## 3.2 Setup

**Model** We derive representations from the 7[th] layer of mBERT. The language-agnostic function $f_\phi$ is parameterized by a 2-layer perceptron with $h$ hidden units that projects the features derived from mBERT to an $m$-dimensional space. We set $h = 256$ and $m = 32$ for word class identification, i.e., part-of-speech (POS) tagging. For grammatical relation, we set $h = 384$ and $m = 64$.

**Data** To investigate the impact of languages involved in meta-training on performance, we examine two distinct settings: i) **M28** with 28 high-resource languages included in meta-training and ii) **M43** with 15 additional languages, which are primarily low-resource ones. Unless otherwise stated, we use English as the source language. The languages and datasets used here is shown in Appendix E.

**Evaluation** We randomly select $N$ sentences from the training set of a target language[5] as the support set, and evaluate the model on its test set in terms of accuracy. We vary $N$ from 0 to 200 to test the number of sentences needed for generalization.

**Baselines** We compare our method with the following baselines: (i) **FT** The mBERT model is fine-tuned with the $N$ available sentences in the target language, with a linear classifier on top of

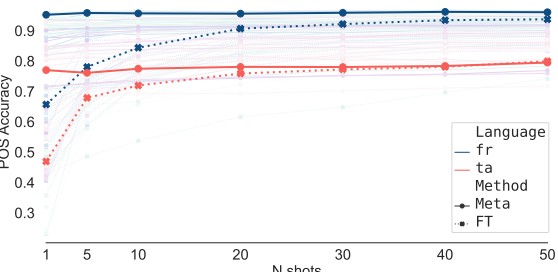

Figure 3: Accuracy of M28 in identifying word classes in languages used during meta-training (depicted by bluish lines) and novel languages encountered during meta-testing (depicted by reddish lines) given $N$ sentences. (The results for the classification of grammatical relations are shown in Appendix B.)

it as the task-specific layer. (ii) **UDapter**, which is a state-of-the-art multilingual parser (Üstün et al., 2022) that extends mBERT with adapters whose parameters are generated based on language embeddings from URIEL (Littell et al., 2017).

## 3.3 Results

**Explicit alignment benefits low-resource languages** Our method achieves competitive results with UDAPTER without any parameter updates to the LLM (Table 1), and even surpasses it for some languages distant from high-resource ones like Marathi (mr) and Warlpiri (wbp). Moreover, our method helps mitigate the performance gap between different languages, as shown in the reduced standard deviation among languages.

**Meta-learning supports efficient alignment** Comparing M28 and M43, we observe that, by incorporating low-resource languages like Marathi (mr) into meta-training, even with limited data[6], substantial performance gains can be achieved

---

[5] For languages without training sets, we select sentences from their test sets and evaluate on the remaining data.

[6] The training set of Marathi consists of only 373 sentences.

without compromising performance in other languages. This suggests that the inclusion of diverse languages in meta-training is critical for learning to derive conceptual spaces from LLMs. Based on the knowledge meta-learned from various languages, the model efficiently learns to align unseen languages with little supervision and optimizes at about 50 sentences, in contrast to FT (Figure 3), supporting the effectiveness of our method.

## 3.4 Discussion

Our results taps into the potential for aligning systems possessing common structure, which has shown to support generalization even in the absence of explicit supervision (Roads and Love, 2020; Aho et al., 2022). Previous work has suggested that word embeddings in different languages are automatically aligned through joint pretraining (Cao et al., 2019; Conneau et al., 2020). In terms of structural concepts, their correspondence is reflected in the geometry of pretrained LLMs and have been aligned to a certain extent (Chi et al., 2020). However, the alignment is not optimal and can be improved with a few examples, as suggested by Lauscher et al. (2020). Our approach may be expanded to other aspects of language and, by utilizing the knowledge acquired via self-supervision, it is promising to better accommodate the rich linguistic diversity despite the scarcity of labeled data.

## 4 Aligning Conceptual Correspondence during In-Context Learning

In this section, we further explore whether the correspondence between structural concepts can be harnessed for cross-lingual generalization in the in-context learning setting. Specifically, we first show that the cross-lingual syntactic abilities of LLMs can be elicited through in-context learning. We then rely on our method to probe the underlying mechanisms and further enhance the cross-lingual generalization by aligning the structural concepts within different languages.

### 4.1 Method

**Learning cross-lingual structural concepts in context** We focus on the POS tagging task and use the structured prompting method proposed by Blevins et al. (2022) to evaluate the few-shot in-context learning ability of LLMs (Figure 4), where

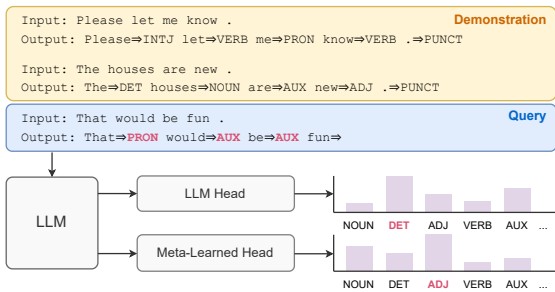

Figure 4: Sequence tagging via structured prompting. The classification of structural concepts is performed in a sequential manner.

the model is given a small number of demonstration examples and then required to label additional sentences in either the same language or a different one. During in-context learning, an LLM is provided with $N$ pairs of sentences and tagged sequences as task demonstrations and a query sentence to be labeled. It is then required to iteratively tag the words. Specifically, given an input sequence $\ell$ with $N$ demonstration examples and the query sentence $S = s_1, \ldots, s_n$, at each time step $t$, the LLM $\mathcal{M}$ encodes $[\ell; s_t]$ and generate the label of $s_t$ with $\hat{c}_t = \arg\max_c P_{\mathcal{M}}(c \mid \ell, s_t)$. The input sequence is then updated with the predicted label $\hat{c}_t$ and the following word $s_{t+1}$ appended to the end of $\ell$.

**Probing underlying mechanisms** Firstly, we take the demonstration examples as query sentences, and evaluate the accuracy of the LLM in classifying these examples. We then investigate whether the representation space contextualized by the demonstrations effectively serves as a conceptual space where samples can be classified based on their distances to prototypes for each concept. We use the $N$ demonstration examples as query sentences and obtain their representations $\mathbf{h}_t$ at time step $t$ for generating the label $\hat{c}_t$, whereby we construct a dataset $\mathcal{D} = \{\mathbf{x}_i, y_i\}_{i=1}^N$ based on $\mathbf{h}_t$ and the gold label $c_t$. We follow the approach in Section 2.1 to probe the extent to which structural concepts can be derived from the contextualized representation space, with the exception that the linear transformation is an identity matrix. Finally, we modify the labels and languages provided in demonstrations to assess whether our results generalize across different settings.

**Meta-learning for better generalization** Our analysis demonstrates that the LLM learns to accurately label the demonstration examples through

in-context learning, but the generalization performance falls short in both monolingual and cross-lingual settings. We thus rely on our meta-learning-based method to improve the LLM's generalization ability. As previously mentioned, we obtain prototypes by utilizing the demonstration examples as query sentences, and then learn to align representations of other query sentences, contextualized by these demonstrations, with the prototypes. This resembles the zero-shot setting in Section 3.1, but we discard the linear mapping applied to the prototypes, as the prototypes themselves are projected into a contextualized representation space and serve as good anchor points. During meta-learning, we introduce varying demonstration examples to construct different training episodes, and the model is directly applied across different contexts for meta-testing.

## 4.2 Setup

**Model**  We use LLaMA-7B as the underlying LLM. The network used for meta-learning resembles the one described in Section 3.2, which is a 2-layer perceptron with a hidden layer of size 512.

**Data**  We employ 24 languages for our experiments, among which 5 languages are used for meta-training. We represent each POS tag with the token corresponding to the surface form of the label defined in UD by default (UPOS), e.g., "NOUN". We investigate three additional settings where the label forms are modified: i) SHFL, which shuffles the surface forms of the labels, ii) PXY, which uses proxy labels where each class is represented by an arbitrary token—we employ capital alphabet letters here, and iii) WORD, which uses words as labels, e.g., "adverb"[7].

**Evaluation**  We randomly select 9 sentences from the training set in a source language as demonstrations, ensuring they cover the label space if possible. For in-context learning, the LLM is evaluated on 50 randomly selected sentences from the test set for each language. We report the average accuracy across 10 runs, where a sample is considered correctly labeled only if the first word the LLM generates after seeing the delimiter matches the form of the gold label. In terms of the probing and meta-learning experiments, we focus on the setting where

---

[7]The languages and datasets are listed in Appendix E, and the detailed label forms can be found in Appendix C.

| | UPOS | SHFL | PXY | WORD |
|------|------|------|------|------|
| ICL | 99.4 | 99.4 | 99.9 | 99.2 |
| Proto | 92.7 | 93.5 | 99.3 | 96.1 |

Table 2: Accuracy in POS tagging in English (en) for demonstrations taken as query sentences. ICL denotes the performance achieved through in-context learning, and Proto for classification based on prototypes computed based on our method.

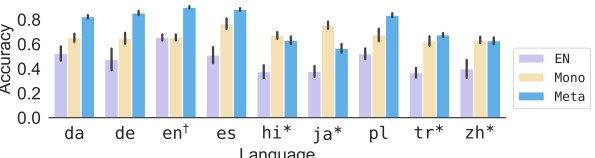

Figure 5: The few-shot generalization performance on POS tagging in the monolingual (MONO) and cross-lingual (EN) in-context learning settings, with English as the source language. META denotes our meta-learning-based method. Error bars represent the standard deviation calculated from 10 runs. Languages marked with "∗" are not included in the pretraining corpus. "†" indicates that the language is involved in meta-training for META. (The results for all languages are presented in Appendix C.2.)

the demonstrations provided are in English and perform 10 runs for each language with 50 query sentences randomly sampled from the training set of UD. The evaluation set for each language consists of 10 runs under similar settings, with the exception that the query sentences are sampled from the test set of UD.

## 4.3 Results

**LLM successfully learns to label the demonstration examples, but with limited generalization abilities**  Table 2 shows that the LLM is able to accurately label demonstration examples, regardless of changes in the label forms. Moreover, the contextualized representations of the demonstrations can be effectively classified based on the prototypes, indicating a good conceptual space for them. However, the performance significantly decreases when generalizing to unseen sentences, and the cross-lingual generalization performance can be even worse (Figure 5).

**Aligning with demonstrations supports generalization**  Through learning to align with prototypes derived from a few demonstration examples, our method achieves remarkable gains in generalization for both monolingual and cross-lingual scenar-

ios (Figure 5). Extending the inclusion of diverse languages in our method holds promise for further improving cross-lingual generalization, particularly for languages that are not well represented in the meta-training languages, like Japanese (ja). These findings support the effectiveness of our method even in the face of variations introduced by changes in demonstrations, and suggest that better generalization performance can be achieved through explicit alignment.

## 4.4 Discussion

While in-context learning has shown an efficient way to leverage LLMs for various downstream tasks and enables few-shot generalization (Brown et al., 2020; Winata et al., 2022), the underlying mechanisms remain unclear. Previous research has suggested that prompting can be regarded as probing knowledge from LLMs (Li et al., 2022a; Blevins et al., 2022; Alivanistos et al., 2022), but the performance is sensitive to prompt engineering, including factors such as the label space and the input distribution (Zhao et al., 2021; Lu et al., 2022; Min et al., 2022; Mishra et al., 2022). We here investigate the representation space of LLMs and find the demonstrations are effectively learned by them despite changes in the label forms. These demonstrations establishes a conceptual space with which we may align different samples. Our findings are in line with Olsson et al. (2022), suggesting that LLMs learn to match the patterns in the context, and offer insights into improving the generalization abilities of LLMs beyond prompt design.

## 5 Related Work

**Probing linguistic knowledge** Pretrained LLMs have shown able to induce sophisticated linguistic knowledge via self-supervision (Manning et al., 2020; Linzen and Baroni, 2021). As evidenced by probing analyses, structural information including word classes, grammatical relations and syntactic parse trees can be decoded from their representations to a remarkable extent (Conneau et al., 2018; Blevins et al., 2018; Liu et al., 2019; Tenney et al., 2019; Clark et al., 2019; Hewitt and Manning, 2019; Eisape et al., 2022). Expanded to the multilingual setting, these models automatically capture nuanced similarities and differences between languages (Chi et al., 2020; Papadimitriou et al., 2021; Singh et al., 2019;

Bjerva and Augenstein, 2021; Xu et al., 2022), enabling efficient zero-shot cross-lingual transfer. We here further explore whether the correspondence between structural concepts are reflected in LLMs' representation space, which could potentially be harnessed for better generalization.

**Cross-lingual generalization** While multilingual LLMs are capable of zero-shot cross-lingual transfer across various tasks (Pires et al., 2019; Wu and Dredze, 2019; Winata et al., 2021), the performance is sensitive to linguistic diversity. Efforts have been made to facilitate generalization to low-resource languages by learning proper information sharing (Ammar et al., 2016; Üstün et al., 2020; Nooralahzadeh et al., 2020) and optimizing data selection (Ponti et al., 2018; Lin et al., 2019; Glavaš and Vulić, 2021), but problems remain when it comes to outlier languages for which there is no high-resource related ones (Blasi et al., 2022). Another line of work strives to overcome the language barriers by imposing alignment of word embeddings (Lample et al., 2018; Ruder et al., 2019; Schuster et al., 2019; Cao et al., 2019). While the alignment typically ensures that semantically and syntactically similar words are clustered together, it is left implicit whether the underlying conceptual correspondence between languages are properly aligned.

**Meta-learning for generalization** Meta-learning has showcased great success in enabling effective generalization. Through the process of learning to learn, namely, improving learning over multiple learning episodes (Wang et al., 2020; Huisman et al., 2021; Hospedales et al., 2022), it facilitates rapid adaptation to novel contexts with limited data available. Prior work has exploited methods including Model-Agnostic Meta-Learning (Finn et al., 2017), Reptile (Nichol et al., 2018) and Prototypical Networks (Snell et al., 2017) for improved cross-lingual generalization (Ponti et al., 2021; Langedijk et al., 2022; Sherborne and Lapata, 2023; Cattan et al., 2021). Our method is similar to Prototypical Networks, but instead of estimating prototypes with the few available samples for each class, we derive prototypes from a source language and learn to align different languages with them, verifying the conceptual correspondence captured by LLMs.

## 6   Conclusion

We have demonstrated that multilingual LMs are able to induce the correspondence between structural concepts within different languages without any explicit supervision. This knowledge is encoded in their geometry and can be exploited for generalization, whereby we rely on meta-learning to learn to align different languages with minimal examples available. Our approach can be used to evaluate the correspondence between different systems that has been acquired by LLMs, and explicitly leverage them for sample-efficient generalization, suggesting a new path toward measuring and manipulating the knowledge encoded in LLMs. Future research may generalize it to other contexts (e.g., other modalities) to probe the commonalities and differences shared between systems and develop more sophisticated way for alignment and generalization.

## Limitations

Our goal in this work is to measure the underlying conceptual correspondence between languages encoded in LLMs, and leverage it for generalization. While we have demonstrated the effectiveness of our approach, it is only a first step toward the more general goal. The foremost limitation of our approach is that it relies on the comparable concepts defined by linguists and manually created datasets to derive proper features for analyses. Continued research could expand upon it to other tasks where prior knowledge of association is available and explore how different kinds of alignment impact the performance of LMs.

Despite the correspondence, our findings suggest that the alignability as well as the generalization performance still varies across languages, though not as pronounced as in the zero-shot cross-lingual transfer scenario. These variations may be attributed to factors like nuanced cross-linguistic differences, degraded representation space for some low-resource languages and insufficient task-specific data. Further investigation is needed to fully understand the cause of these disparities and how they can be reduced to improve the generalization abilities of pretrained LMs, e.g., how to improve the representation spaces of individual languages.

## Acknowledgements

The authors would like to thank the anonymous reviewers for their helpful comments. This work was partially funded by National Natural Science Foundation of China (No.62076069, 61976056).

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

# A   Additional Materials for Correspondence between Structural Concepts in Transformers

## A.1   Identification of Structural Concepts Based on Prototypes

**Representations of structural concepts**   Given an input sequence $\ell$ of $n$ tokens $w_{1:n}^\ell$, an LLM

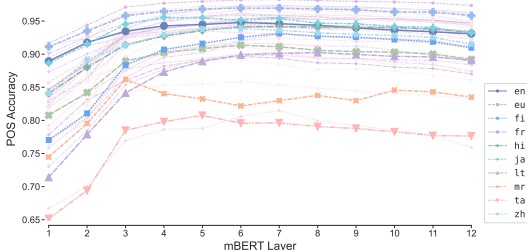

Figure 6: Accuracy in identifying word classes of different languages across different layers of mBERT.

produces contextual representations $\mathbf{h}_{1:n}^\ell$ for each of the token $w_i^\ell$ $(i = 1, \ldots, n)$. We take the representation $\mathbf{h}_i^\ell$ corresponding to the word $w_i^\ell$ as the feature of its word class. The feature of the grammatical relation between a head-dependent pair of words $\left(w_{\text{head}}^\ell, w_{\text{dep}}^\ell\right)$ is given by the difference between their representations:

$$\mathbf{r}_{(\text{head,dep})}^\ell = \mathbf{h}_{\text{head}}^\ell - \mathbf{h}_{\text{dep}}^\ell, \qquad (4)$$

akin to previous works (Hewitt and Manning, 2019; Chi et al., 2020; Xu et al., 2022). The features and corresponding labels constitute the datasets $\mathcal{D} = \left\{\mathbf{x}_i, y_i\right\}_{i=1}^N$, whereby we derive each structural concept $k$ and measure the alignablity between languages (Section 2.1).

**Validation of method**   The middle layers of BERT-like models have shown most effective in encoding syntactic information (Hewitt and Manning, 2019; Chi et al., 2020). We validate this applies to our setting through probing the different layers of mBERT. Besides, to ensure that our method reflects the information about structural concepts encoded in the representation space, we compare the performance in classification with the following baselines: 1) **LAYER0**, the 0th layer of mBERT, where no contextual information is given; and 2) **RAND**, a model shares the same architecture as mBERT with its weights randomized[8]. For these experiments, we set the maximum rank of the probe model to 768 to maximally extract relevant information encoded in the representations.

**Results**   The $7^{\text{th}}$ and $8^{\text{th}}$ layers of the model are most effective in encoding the grammatical relations (Figure 7). For word classes, performance is relatively consistent from the $3^{\text{rd}}$ to $9^{\text{th}}$ layer except for some low-resource languages like

---

[8]As the performance of RAND is approximately equal across different layers, we consistently select the $7^{\text{th}}$ layer for our analysis.

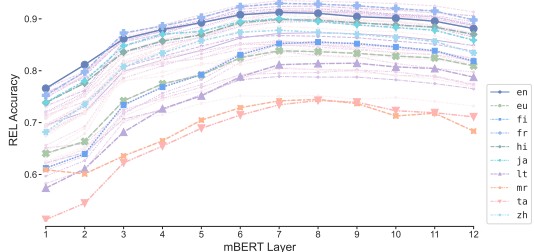

Figure 7: Accuracy in identifying grammatical relations of different languages across different layers of mBERT.

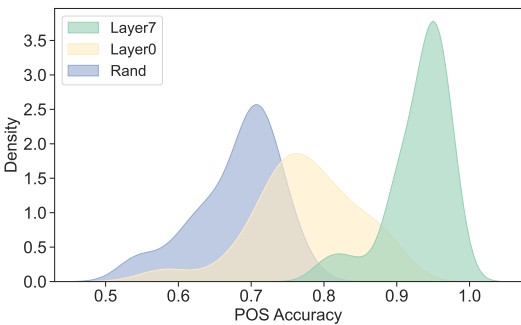

Figure 8: The distribution of the accuracy in deriving word classes from the $7^{th}$ layer of mBERT, along with two baselines. The x-axis denotes the accuracy, of which the distribution is derived from the results in 43 languages. The Wilcoxon test shows that the $7^{th}$ layer exhibits a significantly higher performance ($W = 0.0$, $p = 2.27 \times 10^{-13}$).

Marathi (mr) and Tamil (ta) (Figure 6). We thus take the $7^{th}$ layer for our experiments. The comparison with the two baselines (Figure 8 and Figure 9) supports the efficacy of our method in deriving prototypes of structural concepts while reflecting the geometry of LMs.

Moreover, we also observe disparities between languages reflected in the classification of structural concepts. As shown in Figure 6 and Figure 7, the performance on low-resource languages like Marathi and Tamil consistently lags behind, indicating insufficient representation of these languages in LLMs.

## A.2 Alignability between Structural Concepts in Different Languages

**Details of evaluation** We use RSA and Procrustes analysis[9] (PA) to measure the alignability between structural concepts in different languages. The RSA between two languages is evaluated through the Spearman's rank correlation between

---
[9] https://docs.scipy.org/doc/scipy/reference/generated/scipy.spatial.procrustes.html

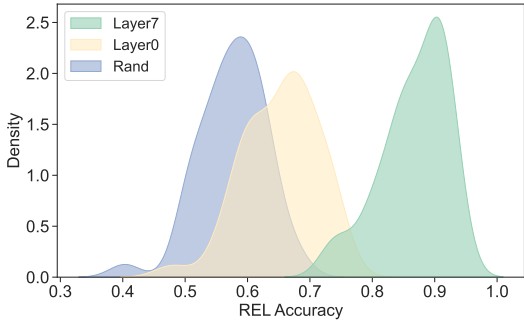

Figure 9: The distribution of the accuracy in deriving grammatical relations from the $7^{th}$ layer of mBERT, along with two baselines. The x-axis denotes the accuracy, of which the distribution is derived from the results in 43 languages. The Wilcoxon test shows that the $7^{th}$ layer exhibits a significantly higher performance ($W = 0.0$, $p = 2.27 \times 10^{-13}$).

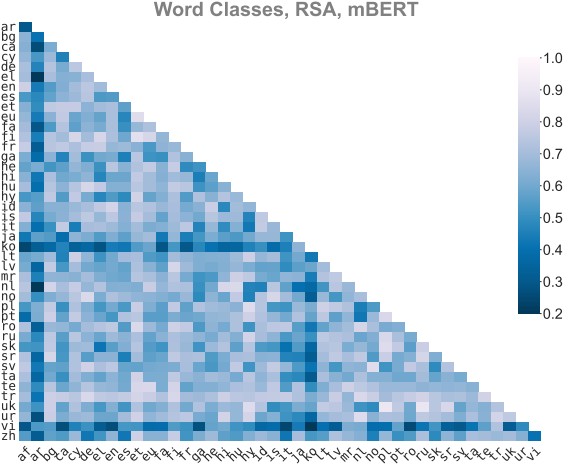

Figure 10: The alignability between word classes within different languages in mBERT measured by RSA.

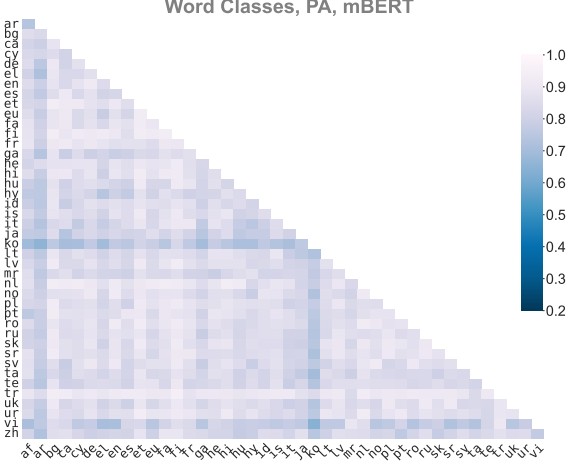

Figure 11: The alignability between word classes within different languages in mBERT measured by Procrustes analysis.

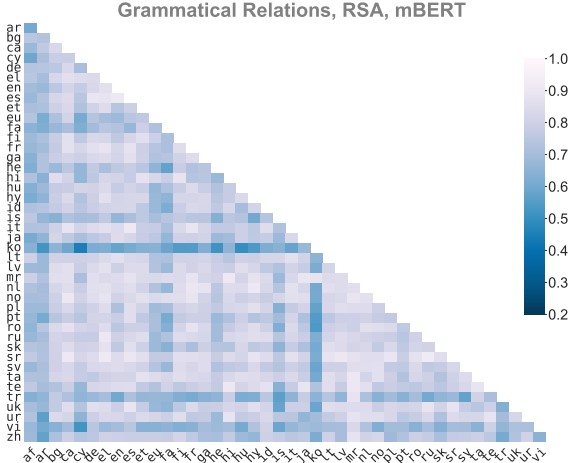

Figure 12: The alignability between grammatical relations within different languages in mBERT measured by RSA.

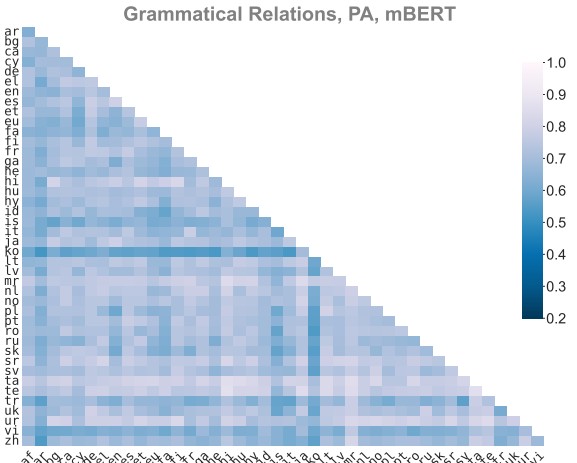

Figure 13: The alignability between grammatical relations within different languages in mBERT measured by Procrustes analysis.

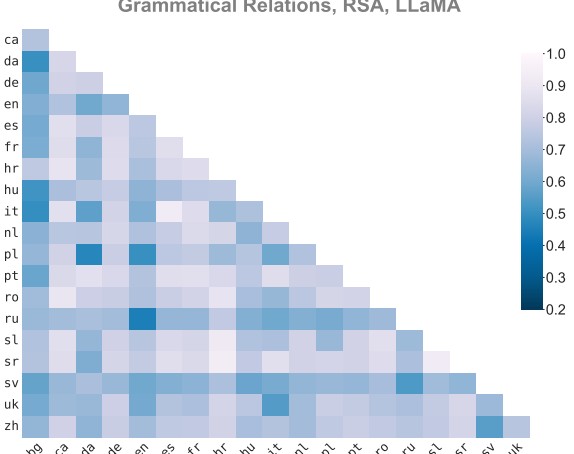

Figure 14: The alignability between grammatical relations within different languages in LLaMA measured by RSA.

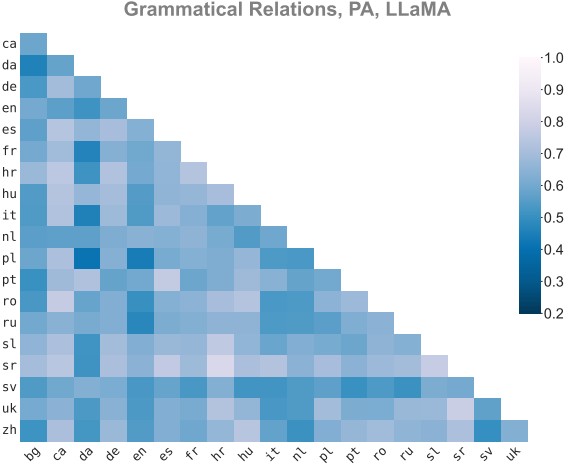

Figure 15: The alignability between grammatical relations within different languages in LLaMA measured by Procrustes analysis.

the lower diagonal portion of their dissimilarity matrices. The fitness of the linear transformation derived from PA is evaluated through the average proportion of explained variance[10].

**Details of baselines** Given the prototypes for $K$ structural concepts derived from two languages $L_1$ and $L_2$, we construct the following three baselines: (i) **RP**, which randomly swaps each prototypes for another in one of the two languages; (ii) **RC**, where we randomly select a sample of each concept instead of their prototypes in one language; (iii) **RS**, where we randomly select $K$ samples in one language. Each baseline thus creates a different mapping between two languages, and we test 100

[10]https://scikit-learn.org/stable/modules/generated/sklearn.metrics.r2_score.html.

random mappings per baseline. We employ the Wilcoxon test to assess whether the alignability between a language pair in LLMs, computed based on our method, is significantly higher than these baselines.

**Results** The alignability between all 43 languages in mBERT with regard to word classes is shown in Figure 10 (RSA) and Figure 11 (PA). Figure 12 (RSA) and Figure 13 (PA) show the results for grammatical relations in mBERT. In terms of LLaMA, the results for word classes are shown in Section 2.3, and the results for grammatical relations are depicted in Figure 14 (RSA) and Figure 15 (PA). The alignability in mBERT and LLaMA is both significantly higher than the baselines, with $p < 0.001$ for almost all

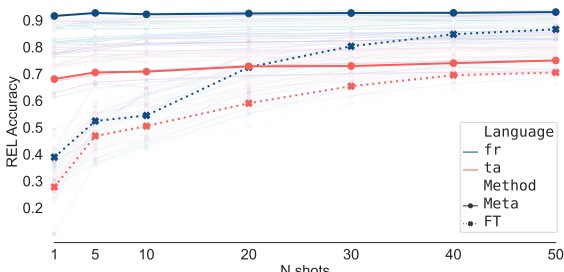

Figure 16: Accuracy of M28 in identifying grammatical relations in languages used during meta-training (depicted by bluish lines) and novel languages encountered during meta-testing (depicted by reddish lines) given $N$ sentences.

language pairs. The only exception arises when comparing the alignability with RC, where the samples are randomly taken from the same class (Table 3).

# B  Additional Materials for Aligning Conceptual Correspondence for Cross-Lingual Generalization

For POS tagging, Table 6 shows our results on 30 low-resource languages compared with UDAPTER. Results on other languages are shown in Table 7. The results for the classification of grammatical relations are shown in Figure 16, Table 8 and Table 9.

Additionally, we present the average accuracy and standard deviation for both low-resource and high-resource languages, as shown in Table 4 and Table 5. With an increasing number of available examples, our approach demonstrates consistent improvements and helps mitigate the performance gap across diverse languages.

# C  Additional Materials for Aligning Conceptual Correspondence during In-Context Learning

## C.1  Experimental Details

**Label forms**  The label forms used in our experiments are as follows:

- UPOS: ADJ, ADP, ADV, AUX, CCONJ, DET, INTJ, NOUN, NUM, PART, PRON, PROPN, PUNCT, SCONJ, SYM, VERB, X.

- SHFL: PUNCT, DET, AUX, ADJ, PRON, X, PART, CCONJ, INTJ, NUM, SCONJ, ADV, SYM, VERB, PROPN, ADP, NOUN.

- PXY: A, B, C, D, E, F, G, H, I, J, K, L, M, N, O, P, Q.

- WORD: adjective, adposition, adverb, auxiliary, coordinating_conjunction, determiner, interjection, noun, numeral, particle, pronoun, proper_noun, punctuation, subordinating_conjunction, symbol, verb, other.

**Languages for meta-training**  We employ five languages for meta-training, including bg, en, fi, fr, ru. The other languages and datasets used in our experiments are shown in Appendix E.

## C.2  Full Results

Figure 17 presents the full results for our meta-learning-based methods across 24 languages.

# D  Implementation Details

**Large language models**  We use Multilingual BERT (bert-base-multilingual-cased)[11] and LLaMA-7B (llama-7b-hf)[12] for all our experiments.

**Deriving structural concepts from LLMs**  For all our experiments that derive structural concepts from LLMs through a linear transformation, we train the linear probe with a batch size of 8 and a max sequence length of 128 for 20 epochs, and validate it at the end of each epoch. We select the model performing the best on the development set. We use the Adam optimizer with $\beta_1 = 0.9$, $\beta_2 = 0.999$, and a weight decay of $1 \times 10^{-6}$. The learning rate is set to $1 \times 10^{-4}$.

**Learning to align conceptual correspondence**  Our meta-learning-based method follows the procedure described above to derive prototypes for each concept. Subsequently, the networks are trained for 50 epochs, with a maximum sequence length of 128. During meta-training, given $m$ languages, each epoch consists of $m \times 50$ training episodes. These episodes are constructed using $N$ labeled sentences as the support set and 30 labeled sentences as the query set. The method we use here bears resemblance to Li et al. (2022b). The parameters of the network are optimized through the Adam optimizer, with $\beta_1 = 0.9$, $\beta_2 = 0.999$, and a weight decay of $1 \times 10^{-4}$. The learning

---

[11]https://huggingface.co/bert-base-multilingual-cased.
[12]https://huggingface.co/decapoda-research/llama-7b-hf.

| Model | $L_1$ | $L_2$ | Concept | Measure | Baseline | $p$-value |
|-------|-------|-------|---------|---------|----------|-----------|
| mBERT | ar | el | POS | RSA | RC | $p = 0.040 < 0.05$ |
| mBERT | ar | nl | POS | RSA | RC | $p = 0.030 < 0.05$ |
| mBERT | ko | de | POS | RSA | RC | $p = 0.009 < 0.01$ |
| mBERT | ko | de | POS | RSA | RC | $p = 0.009 < 0.01$ |
| mBERT | ar | mr | POS | PA | RC | $p = 0.130$ |
| mBERT | en | vi | POS | PA | RC | $p = 0.002 < 0.005$ |
| mBERT | ko | vi | POS | PA | RC | $p = 0.616$ |
| LLaMA | bg | de | POS | PA | RC | $p = 0.027 < 0.05$ |
| LLaMA | bg | fr | POS | PA | RC | $p = 0.007 < 0.01$ |

Table 3: Exceptions where the alignability between the language pair exceeds a significance threshold of $p < 0.001$.

|         | LR-AVG | LR-STD | HR-AVG | HR-STD | AVG  | STD    |
|---------|--------|--------|--------|--------|------|--------|
| UDAPTER | 58.4   | 0.2135 | 97.0   | 0.0113 | 70.0 | 0.2516 |
| M28-0   | 56.6   | 0.2118 | 90.8   | 0.0480 | 66.9 | 0.2379 |
| M28-10  | 59.0   | 0.2075 | 89.2   | 0.0581 | 68.1 | 0.2245 |
| M28-30  | 61.5   | 0.1874 | 90.7   | 0.0507 | 70.3 | 0.2080 |
| M28-50  | 62.5   | 0.1834 | 91.0   | 0.0478 | 71.1 | 0.2032 |
| M43-0   | 57.9   | 0.2280 | 90.9   | 0.0468 | 67.9 | 0.2446 |
| M43-10  | 60.8   | 0.2111 | 89.6   | 0.0596 | 69.5 | 0.2229 |
| M43-30  | 63.0   | 0.1942 | 90.6   | 0.0541 | 71.4 | 0.2079 |
| M43-50  | 64.4   | 0.1869 | 91.1   | 0.0495 | 72.5 | 0.2003 |

Table 4: The average accuracy (AVG) and standard deviation (STD) with regard to POS tagging measured across the languages used in UDAPTER, where the low-resource languages (LR) are the 30 languages listed in Table 6 and the high-resource languages (HR) include: ar, en, eu, fi, he, hi, it, ja, ko, ru, sv, tr and zh. AVG and STD are computed over all languages. While our method lags behind UDAPTER in terms of high-resource languages, its demonstrates an increasing performance on low-resource ones when provided with additional examples. Moreover, the gap between different languages becomes smaller, especially for M28, which does not involve any low-resource languages in meta-training.

rate is set to $5 \times 10^{-5}$. The hidden layer dropout probability is 0.33.

Note that as the conceptual correspondence holds across languages for concepts with (Section 2.3) for concepts with more than (or equal to) 20 samples, concepts with fewer than 20 samples in either the source or target languages are excluded during meta-training. During meta-testing, examples belonging to these categories are considered as misclassified.

**Aligning conceptual correspondence during in-context learning** For meta-learning during in-context learning, our networks are trained for 100 epochs with each consists of $m \times 10$ episodes, where $m = 5$ is the number of languages involved in training. The parameters of the network are optimized through the Adam optimizer, with $\beta_1 = 0.9$, $\beta_2 = 0.999$, and a weight decay of $1 \times 10^{-4}$. The learning rate is set to $5 \times 10^{-4}$. The hidden

layer dropout probability is 0.33.

## E Data

The data used in all our experiments is from UD v2.10 (Zeman et al., 2022)[13]. We follow the split of training, development and test set in it.

**Data for measuring correspondence between structural concepts within different languages** In terms of measuring the correspondence between structural concepts within different languages (Section 2), we use 28 languages for mBERT, including af, ar, bg, ca, cy, de, el, en, es, et, eu, fa, fi, fr, ga, he, hi, hu, hy, id, is, it, ja, ko, lt, lv, mr, nl, no, pl, pt, ro, ru, sk, sr, sv, ta, te, tr, uk, ur, vi and zh. For LLaMA, we test on 20 languages: bg, ca, da, de, en, es, fr, hr, hu, it, nl, pl, pt, ro, ru, sl,

---
[13]https://lindat.mff.cuni.cz/repository/xmlui/handle/11234/1-4758.

|        | LR-AVG | LR-STD | HR-AVG | HR-STD | AVG  | STD    |
|--------|--------|--------|--------|--------|------|--------|
| M28-0  | 53.1   | 0.1870 | 84.7   | 0.0525 | 62.6 | 0.2151 |
| M28-10 | 57.6   | 0.1648 | 83.2   | 0.0667 | 65.3 | 0.1848 |
| M28-30 | 59.9   | 0.1609 | 84.9   | 0.0610 | 67.4 | 0.1802 |
| M28-50 | 60.8   | 0.1597 | 85.6   | 0.0564 | 68.3 | 0.1780 |
| M43-0  | 52.5   | 0.2080 | 81.4   | 0.0717 | 61.3 | 0.2222 |
| M43-10 | 58.6   | 0.1739 | 82.6   | 0.0741 | 65.8 | 0.1869 |
| M43-30 | 60.5   | 0.1681 | 84.6   | 0.0622 | 67.8 | 0.1822 |
| M43-50 | 61.7   | 0.1662 | 85.7   | 0.0559 | 68.9 | 0.1799 |

Table 5: The average accuracy (AVG) and standard deviation (STD) with regard to the classification of grammatical relations measured across different languages, where the low-resource languages (LR) are the 30 languages listed in Table 6 and the high-resource languages (HR) include: ar, en, eu, fi, he, hi, it, ja, ko, ru, sv, tr and zh. AVG and STD are computed over all languages. Our method demonstrates an increasing performance on low-resource ones when provided with additional examples. Moreover, the gap between different languages becomes smaller, especially for M28, which does not involve any low-resource languages in meta-training.

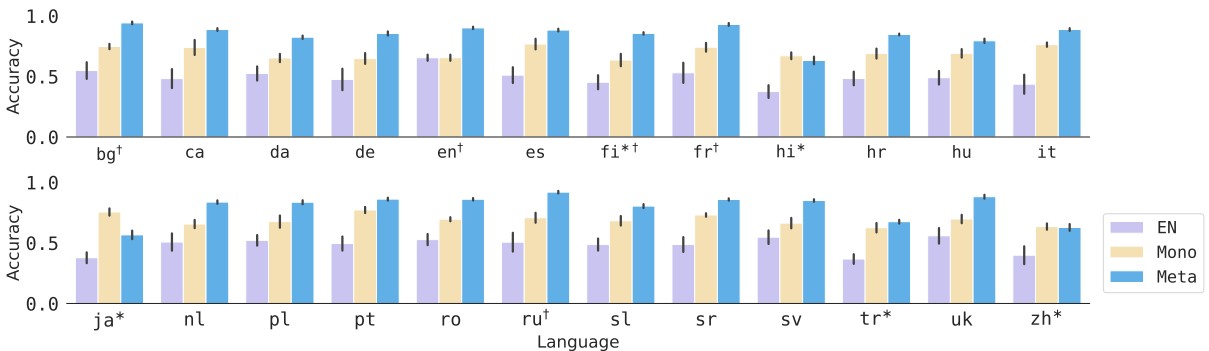

Figure 17: The few-shot generalization performance on POS tagging in the monolingual (MONO) and cross-lingual (EN) in-context learning settings, with English as the source language. META denotes our meta-learning-based method. Error bars represent the standard deviation calculated from 10 runs. Languages marked with "∗" are not included in the pretraining corpus. "†" indicates that the language is involved in meta-training for META.

sr, sv, uk and zh.

**Data for aligning conceptual correspondence for cross-lingual generalization** The 28 languages involved in the meta-training of **M28** are: ar, bg, ca, de, es, et, eu, fa, fi, fr, he, hi, is, it, ja, ko, lv, nl, no, pl, pt, ro, ru, sk, sv, tr, uk and zh. For **M43**, the 15 additional languages used for meta-training include: af, cy, el, en, ga, hu, hy, id, lt, mr, sr, ta, te, ur and vi. Both are evaluated on a total of 72 languages, which involves the 43 languages for meta-training and another 29 languages: aii, akk, am, be, bho, bm, br, bxr, cs, da, fo, gsw, gun, hsb, kk, kmr, koi, kpv, krl, mdf, myv, olo, pcm, sa, sl, tl, wbp, yo and yue.

**Data for aligning conceptual correspondence during in-context learning** The experiments with in-context learning encompass 24 languages, including bg, ca, da, de, en, es, fr, hr, hu, it, nl, pl,

pt, ro, ru, sl, sr, sv, uk and zh, among which zh, fi, hi, ja, tr are not included in the pretraining corpus.

**Detailed language information** Table 10 and Table 11 provide detailed information on the languages utilized in our experiments, including their respective language codes, sizes of the UD datasets, and language families.

| | aii* | akk* | am* | be | bho* | bm* | br* | bxr* | cy† | fo* | gsw* | gun* | hsb* | kk | kmr* |
|---|---|---|---|---|---|---|---|---|---|---|---|---|---|---|---|
| UDAPTER | 14.8 | 20.4 | 10.9 | 96.9 | 63.1 | 35.8 | 72.2 | 65.6 | 69.7 | 79.6 | 65.9 | 36.3 | 78.8 | 83.4 | 48.8 |
| M28-0 | 14.6 | 21.9 | 20.9 | 90.3 | 62.5 | 32.7 | 75.4 | 60.1 | 69.9 | 84.6 | 64.7 | 12.9 | 73.5 | 77.2 | 42.4 |
| M28-10 | 22.1 | 30.0 | 14.9 | 89.3 | 63.3 | 35.7 | 79.4 | 60.5 | 73.4 | 85.1 | 68.8 | 14.9 | 75.3 | 79.0 | 48.1 |
| M28-30 | 24.5 | 34.1 | 23.8 | 91.0 | 64.2 | 37.9 | 81.3 | 59.7 | 75.2 | 85.6 | 70.5 | 31.6 | 76.0 | 79.8 | 49.2 |
| M28-50 | 24.3 | 36.5 | 29.7 | 90.7 | 64.6 | 40.7 | 81.1 | 60.0 | 76.9 | 86.3 | 70.6 | 29.0 | 75.8 | 79.8 | 48.6 |
| M28-100 | - | - | 44.4 | 91.8 | 65.9 | 43.8 | 82.7 | 59.6 | 77.6 | 86.9 | - | 33.9 | 75.8 | 79.6 | 49.1 |
| M28-200 | - | - | 49.0 | 92.6 | 66.2 | 48.9 | 83.4 | 59.8 | 78.6 | 87.0 | - | 36.5 | 75.8 | 79.6 | 49.1 |
| M43-0 | 19.0 | 21.0 | 7.5 | 90.3 | 62.0 | 33.6 | 73.9 | 59.5 | 88.5 | 83.8 | 63.4 | 16.6 | 73.9 | 77.7 | 43.1 |
| M43-10 | 30.9 | 28.1 | 14.1 | 89.5 | 62.4 | 36.7 | 80.3 | 59.8 | 87.3 | 84.9 | 69.0 | 17.7 | 74.6 | 80.2 | 46.1 |
| M43-30 | 27.8 | 35.0 | 32.0 | 91.1 | 63.7 | 40.0 | 81.2 | 59.4 | 88.1 | 85.2 | 70.0 | 24.8 | 75.6 | 80.2 | 47.9 |
| M43-50 | 26.9 | 37.6 | 41.7 | 91.6 | 64.8 | 42.2 | 82.3 | 59.2 | 88.9 | 85.7 | 72.2 | 29.8 | 75.5 | 80.3 | 48.5 |
| M43-100 | - | - | 39.5 | 91.9 | 65.2 | 45.2 | 83.1 | 59.6 | 89.4 | 86.5 | - | 32.4 | 75.5 | 80.1 | 47.0 |
| M43-200 | - | - | 44.0 | 92.7 | 65.7 | 49.8 | 83.9 | 59.5 | 89.6 | 86.7 | - | 34.1 | 75.5 | 80.1 | 48.1 |

| | koi* | kpv* | krl* | mdf* | mr† | myv* | olo* | pcm* | sa* | ta† | te† | tl | wbp* | yo | yue* |
|---|---|---|---|---|---|---|---|---|---|---|---|---|---|---|---|
| UDAPTER | 48.9 | 36.7 | 78.3 | 54.7 | 66.5 | 52.8 | 76.6 | 54.7 | 42.2 | 70.3 | 84.2 | 78.4 | 34.1 | 63.7 | 66.3 |
| M28-0 | 44.8 | 36.8 | 72.7 | 50.4 | 79.3 | 51.3 | 67.0 | 37.7 | 45.2 | 73.1 | 78.2 | 73.8 | 51.6 | 61.0 | 71.1 |
| M28-10 | 46.6 | 39.3 | 73.1 | 51.1 | 80.1 | 51.6 | 66.3 | 43.8 | 49.4 | 77.5 | 80.2 | 80.5 | 54.6 | 63.1 | 72.8 |
| M28-30 | 48.8 | 41.1 | 75.0 | 52.1 | 80.3 | 51.9 | 66.0 | 55.5 | 50.8 | 78.0 | 81.4 | 82.0 | 59.2 | 64.9 | 73.8 |
| M28-50 | 51.0 | 41.7 | 76.7 | 51.7 | 79.0 | 52.7 | 66.0 | 64.3 | 52.0 | 79.5 | 80.3 | 83.7 | 58.8 | 67.0 | 75.3 |
| M28-100 | - | 43.1 | 76.5 | 52.4 | 81.6 | 52.6 | 66.3 | 71.3 | 51.3 | 79.5 | 82.1 | 83.3 | - | 68.0 | 75.9 |
| M28-200 | - | 45.8 | 79.5 | 52.5 | 81.6 | 53.7 | 66.3 | 76.3 | 55.2 | 79.7 | 82.1 | - | - | 70.7 | 77.1 |
| M43-0 | 47.4 | 36.6 | 71.8 | 49.9 | 87.5 | 51.3 | 65.7 | 47.3 | 46.0 | 81.3 | 90.4 | 70.0 | 49.3 | 58.0 | 70.4 |
| M43-10 | 51.9 | 41.7 | 72.5 | 50.3 | 86.7 | 51.5 | 65.9 | 54.5 | 48.7 | 81.9 | 90.0 | 78.6 | 53.9 | 62.2 | 72.0 |
| M43-30 | 53.7 | 42.6 | 75.5 | 52.6 | 88.6 | 51.9 | 65.2 | 65.7 | 48.9 | 81.2 | 89.2 | 82.0 | 54.2 | 63.9 | 74.4 |
| M43-50 | 53.2 | 42.8 | 76.0 | 52.2 | 88.6 | 51.6 | 65.0 | 70.6 | 49.8 | 81.5 | 90.0 | 82.8 | 58.8 | 66.7 | 75.2 |
| M43-100 | - | 43.2 | 76.3 | 52.9 | 88.8 | 52.4 | 65.2 | 74.2 | 51.1 | 81.4 | 90.7 | 83.3 | - | 67.6 | 76.6 |
| M43-200 | - | 44.4 | 78.7 | 53.7 | 88.8 | 53.7 | 65.2 | 77.2 | 54.0 | 82.0 | 90.3 | - | - | 71.2 | 77.8 |

Table 6: The zero-shot and few-shot generalization performance on POS tagging for 30 low-resource languages. Languages marked with "∗" are not included in the pretraining corpus. "†" indicates that the language is involved in meta-training for M43. The UD dataset for some languages comprises less than 100 or 200 sentences, and the few-shot performance for these languages is left unspecified.

| | af† | ar | bg | ca | cs‡ | da‡ | de | el | en† | es | et | eu | fa | fi | fr |
|---|---|---|---|---|---|---|---|---|---|---|---|---|---|---|---|
| UDAPTER | - | 96.8 | - | - | - | - | - | - | 97.0 | - | - | 95.7 | - | 97.3 | - |
| M28-0 | 89.8 | 93.0 | 96.2 | 95.8 | 92.1 | 89.5 | 93.3 | 88.4 | 84.7 | 94.4 | 91.0 | 90.0 | 92.8 | 92.0 | 95.4 |
| M28-10 | 91.3 | 89.5 | 96.2 | 96.0 | 92.6 | 91.1 | 92.9 | 90.1 | 86.2 | 93.9 | 91.0 | 88.7 | 90.5 | 91.5 | 95.7 |
| M28-30 | 91.6 | 92.5 | 96.3 | 96.5 | 93.4 | 91.5 | 92.9 | 91.2 | 88.2 | 94.5 | 91.5 | 89.4 | 91.5 | 92.2 | 95.9 |
| M28-50 | 91.8 | 93.5 | 96.6 | 96.7 | 94.0 | 92.4 | 93.3 | 91.6 | 87.8 | 94.7 | 91.7 | 90.0 | 93.1 | 92.1 | 96.1 |
| M28-100 | 92.8 | 93.8 | 96.7 | 96.9 | 94.3 | 92.8 | 93.4 | 91.9 | 89.0 | 94.9 | 92.0 | 90.2 | 93.2 | 92.5 | 96.1 |
| M28-200 | 93.6 | 93.8 | 96.8 | 97.1 | 94.5 | 92.9 | 93.8 | 92.7 | 89.4 | 95.1 | 92.2 | 90.4 | 93.4 | 92.5 | 96.2 |
| M43-0 | 93.4 | 92.5 | 96.0 | 95.6 | 92.4 | 89.4 | 93.0 | 94.6 | 91.9 | 94.1 | 90.5 | 89.4 | 92.4 | 91.4 | 95.2 |
| M43-10 | 93.2 | 89.1 | 95.5 | 95.8 | 93.2 | 90.6 | 93.0 | 94.0 | 91.0 | 94.2 | 91.0 | 89.1 | 91.3 | 91.1 | 95.4 |
| M43-30 | 94.3 | 92.3 | 96.1 | 96.4 | 93.6 | 91.8 | 93.0 | 94.4 | 92.0 | 94.8 | 90.9 | 89.3 | 92.3 | 91.8 | 95.5 |
| M43-50 | 95.0 | 92.8 | 96.3 | 96.5 | 93.9 | 92.3 | 93.4 | 95.0 | 92.5 | 94.5 | 91.4 | 89.6 | 92.4 | 91.8 | 95.7 |
| M43-100 | 95.4 | 93.3 | 96.5 | 96.6 | 94.5 | 92.6 | 93.5 | 95.3 | 92.7 | 94.8 | 91.8 | 89.7 | 92.8 | 92.2 | 95.8 |
| M43-200 | 95.7 | 93.3 | 96.8 | 97.1 | 94.6 | 92.9 | 93.7 | 95.6 | 93.0 | 95.2 | 92.0 | 90.0 | 93.2 | 92.1 | 96.1 |

| | ga† | he | hi | hu† | hy† | id† | is | it | ja | ko | lt† | lv | nl | no | pl |
|---|---|---|---|---|---|---|---|---|---|---|---|---|---|---|---|
| UDAPTER | - | 97.1 | 97.4 | - | - | - | - | 98.3 | 97.0 | 96.5 | - | - | - | - | - |
| M28-0 | 71.4 | 93.0 | 92.4 | 84.0 | 83.1 | 82.1 | 94.2 | 96.2 | 93.5 | 78.8 | 84.9 | 90.7 | 93.2 | 93.6 | 94.4 |
| M28-10 | 73.9 | 90.2 | 91.4 | 86.3 | 83.9 | 86.7 | 93.3 | 96.5 | 92.5 | 72.9 | 85.0 | 89.6 | 91.9 | 92.6 | 92.7 |
| M28-30 | 75.2 | 91.9 | 92.0 | 88.0 | 84.6 | 87.6 | 93.7 | 96.7 | 93.0 | 76.9 | 86.1 | 90.5 | 92.7 | 93.3 | 93.6 |
| M28-50 | 75.9 | 92.8 | 92.1 | 89.4 | 84.1 | 88.1 | 93.8 | 96.6 | 93.2 | 78.3 | 86.1 | 90.7 | 92.8 | 93.5 | 94.1 |
| M28-100 | 76.9 | 93.2 | 92.9 | 89.6 | 85.3 | 89.2 | 94.5 | 96.8 | 93.8 | 79.6 | 86.6 | 91.0 | 93.7 | 94.0 | 94.9 |
| M28-200 | 77.8 | 93.4 | 93.2 | 90.0 | 85.8 | 89.8 | 94.6 | 96.8 | 93.8 | 79.9 | 86.8 | 91.2 | 94.1 | 94.2 | 95.5 |
| M43-0 | 86.8 | 92.5 | 92.2 | 91.6 | 88.8 | 91.1 | 93.4 | 96.1 | 93.0 | 77.8 | 89.3 | 90.2 | 92.1 | 93.2 | 94.1 |
| M43-10 | 86.6 | 90.9 | 91.2 | 91.4 | 88.4 | 91.5 | 92.7 | 95.9 | 92.3 | 71.8 | 87.4 | 89.8 | 91.8 | 91.0 | 92.8 |
| M43-30 | 86.9 | 91.8 | 91.8 | 92.0 | 89.1 | 91.5 | 93.1 | 96.3 | 93.0 | 75.0 | 89.1 | 90.7 | 93.0 | 92.7 | 93.6 |
| M43-50 | 87.0 | 92.3 | 92.5 | 92.4 | 88.4 | 91.8 | 93.5 | 96.3 | 93.1 | 77.4 | 89.5 | 90.4 | 93.0 | 93.1 | 94.0 |
| M43-100 | 87.5 | 92.9 | 92.8 | 92.2 | 89.3 | 92.1 | 93.8 | 96.6 | 93.2 | 78.6 | 90.2 | 90.6 | 93.8 | 93.6 | 94.7 |
| M43-200 | 87.8 | 93.1 | 93.1 | 92.6 | 89.9 | 92.4 | 94.0 | 96.8 | 93.4 | 79.1 | 90.4 | 91.0 | 94.1 | 94.1 | 95.4 |

| | pt | ro | ru | sk | sl‡ | sr† | sv | tr | uk | ur† | vi† | zh |
|---|---|---|---|---|---|---|---|---|---|---|---|---|
| UDAPTER | - | - | 98.9 | - | - | - | 98.4 | 95.1 | - | - | - | 95.1 |
| M28-0 | 95.2 | 94.9 | 94.7 | 92.6 | 88.5 | 92.2 | 95.1 | 85.0 | 93.8 | 81.3 | 67.1 | 91.5 |
| M28-10 | 95.0 | 94.3 | 93.0 | 91.6 | 90.5 | 94.3 | 95.1 | 82.9 | 92.9 | 82.8 | 69.4 | 89.6 |
| M28-30 | 95.8 | 95.1 | 95.6 | 92.2 | 91.7 | 94.6 | 95.3 | 84.3 | 93.6 | 84.1 | 70.2 | 91.2 |
| M28-50 | 96.1 | 95.1 | 95.0 | 93.4 | 92.0 | 95.2 | 95.2 | 84.6 | 93.8 | 84.2 | 71.7 | 91.7 |
| M28-100 | 96.4 | 95.2 | 95.7 | 93.5 | 92.7 | 95.4 | 95.8 | 85.0 | 94.3 | 85.8 | 75.4 | 92.1 |
| M28-200 | 96.4 | 95.6 | 95.9 | 94.0 | 93.1 | 95.8 | 95.8 | 84.9 | 94.6 | 86.9 | 78.7 | 92.4 |
| M43-0 | 94.9 | 94.4 | 94.7 | 92.5 | 89.5 | 96.2 | 94.9 | 84.4 | 93.5 | 89.6 | 86.3 | 90.6 |
| M43-10 | 94.5 | 94.3 | 94.4 | 91.2 | 90.7 | 95.8 | 95.3 | 83.7 | 93.2 | 88.4 | 79.3 | 89.2 |
| M43-30 | 95.4 | 94.6 | 94.7 | 93.0 | 91.8 | 96.5 | 95.5 | 83.8 | 94.0 | 89.6 | 82.8 | 90.7 |
| M43-50 | 95.7 | 94.9 | 95.6 | 94.1 | 92.7 | 96.4 | 95.2 | 83.8 | 93.7 | 89.7 | 85.3 | 91.1 |
| M43-100 | 95.9 | 95.0 | 96.0 | 94.2 | 92.9 | 97.0 | 95.4 | 84.3 | 94.6 | 90.1 | 87.7 | 91.6 |
| M43-200 | 96.1 | 95.3 | 96.1 | 94.4 | 93.4 | 97.1 | 95.7 | 84.7 | 94.9 | 90.2 | 88.5 | 91.9 |

Table 7: The zero-shot and few-shot generalization performance on POS tagging for 42 other languages. By default, these languages are used for meta-training. Languages marked with "†" is involved in meta-training for M43 but not for M28. "‡" indicates that the language is excluded from meta-training for both M28 and M43.

| | aii* | akk* | am* | be | bho* | bm* | br* | bxr* | cy† | fo* | gsw* | gun* | hsb* | kk | kmr* |
|---|---|---|---|---|---|---|---|---|---|---|---|---|---|---|---|
| M28-0 | 33.3 | 20.8 | 18.2 | 86.6 | 53.2 | 28.7 | 73.4 | 35.6 | 69.0 | 84.5 | 66.2 | 20.9 | 67.3 | 71.1 | 45.6 |
| M28-10 | 43.7 | 29.3 | 47.7 | 86.2 | 58.3 | 37.4 | 77.3 | 36.2 | 71.4 | 85.3 | 68.2 | 27.1 | 68.2 | 72.1 | 48.7 |
| M28-30 | 49.0 | 34.8 | 48.3 | 86.9 | 61.8 | 42.1 | 78.8 | 37.1 | 73.8 | 86.0 | 70.5 | 30.5 | 68.5 | 72.8 | 49.4 |
| M28-50 | 49.7 | 39.9 | 47.8 | 87.3 | 63.3 | 46.0 | 79.5 | 36.9 | 75.1 | 85.9 | 71.7 | 34.0 | 68.3 | 72.8 | 49.9 |
| M28-100 | - | - | 48.5 | 87.8 | 63.9 | 48.6 | 80.2 | 37.0 | 76.4 | 87.2 | - | 36.4 | 68.5 | 72.8 | 49.6 |
| M28-200 | - | - | 48.4 | 88.3 | 65.7 | 53.7 | 81.4 | 37.0 | 77.1 | 87.5 | - | 37.5 | 68.5 | 72.8 | 49.6 |
| M43-0 | 40.0 | 22.0 | 18.4 | 86.0 | 48.0 | 29.6 | 73.5 | 18.1 | 80.5 | 84.5 | 64.0 | 19.8 | 60.9 | 61.7 | 44.0 |
| M43-10 | 46.8 | 28.4 | 46.7 | 87.0 | 55.6 | 37.3 | 77.7 | 36.9 | 78.9 | 85.5 | 69.1 | 26.6 | 67.8 | 72.5 | 49.0 |
| M43-30 | 49.0 | 36.2 | 47.5 | 86.6 | 61.7 | 40.3 | 79.3 | 37.3 | 81.6 | 85.8 | 71.1 | 30.1 | 68.5 | 73.2 | 49.7 |
| M43-50 | 51.2 | 40.0 | 45.9 | 87.5 | 62.9 | 44.0 | 79.9 | 37.3 | 81.9 | 86.2 | 72.6 | 34.7 | 68.4 | 73.2 | 49.7 |
| M43-100 | - | - | 49.9 | 88.4 | 64.4 | 49.8 | 80.0 | 37.3 | 82.8 | 86.4 | - | 36.8 | 68.5 | 73.2 | 49.4 |
| M43-200 | - | - | 50.2 | 88.5 | 65.1 | 53.8 | 80.6 | 37.2 | 83.1 | 87.3 | - | 38.0 | 68.4 | 73.2 | 49.4 |

| | koi* | kpv* | krl* | mdf* | mr† | myv* | olo* | pcm* | sa* | ta† | te† | tl | wbp* | yo | yue* |
|---|---|---|---|---|---|---|---|---|---|---|---|---|---|---|---|
| M28-0 | 48.2 | 38.5 | 64.7 | 49.7 | 71.5 | 48.3 | 44.1 | 32.5 | 44.3 | 65.2 | 75.5 | 72.9 | 53.3 | 54.5 | 55.1 |
| M28-10 | 51.2 | 40.3 | 65.7 | 52.4 | 76.6 | 49.3 | 44.4 | 35.1 | 47.2 | 70.8 | 74.2 | 77.8 | 63.1 | 59.1 | 63.7 |
| M28-30 | 54.1 | 42.2 | 68.6 | 53.1 | 75.8 | 51.2 | 44.4 | 37.2 | 48.1 | 72.9 | 77.1 | 80.2 | 71.6 | 61.9 | 66.9 |
| M28-50 | 54.7 | 42.7 | 70.3 | 52.8 | 76.1 | 51.7 | 44.3 | 37.4 | 45.7 | 75.0 | 79.1 | 81.5 | 73.9 | 63.1 | 68.5 |
| M28-100 | - | 45.7 | 68.4 | 53.5 | 77.9 | 52.6 | 44.2 | 39.4 | 48.1 | 75.0 | 78.5 | 81.6 | - | 64.0 | 70.6 |
| M28-200 | - | 47.0 | 69.5 | 54.2 | 76.9 | 53.7 | 44.2 | 41.6 | 51.7 | 76.0 | 78.5 | - | - | 66.2 | 72.3 |
| M43-0 | 49.3 | 39.1 | 63.2 | 49.4 | 77.4 | 48.3 | 21.8 | 34.6 | 43.4 | 75.6 | 82.5 | 73.4 | 55.6 | 55.2 | 56.9 |
| M43-10 | 52.9 | 40.3 | 66.0 | 51.0 | 78.7 | 48.8 | 43.5 | 37.2 | 47.1 | 75.5 | 81.7 | 79.6 | 66.0 | 59.6 | 63.2 |
| M43-30 | 52.3 | 43.5 | 68.5 | 53.5 | 77.9 | 50.7 | 44.0 | 38.4 | 47.2 | 77.7 | 81.0 | 81.9 | 71.2 | 60.9 | 67.4 |
| M43-50 | 55.2 | 43.5 | 69.4 | 53.3 | 80.3 | 51.4 | 44.0 | 38.8 | 47.6 | 77.1 | 82.9 | 82.0 | 76.1 | 63.3 | 69.2 |
| M43-100 | - | 44.6 | 70.7 | 52.6 | 80.6 | 52.4 | 43.8 | 40.2 | 46.0 | 78.3 | 82.7 | 81.6 | - | 64.7 | 70.8 |
| M43-200 | - | 45.7 | 70.8 | 54.7 | 81.6 | 54.0 | 44.0 | 42.0 | 50.0 | 78.7 | 83.6 | - | - | 66.9 | 71.8 |

Table 8: The zero-shot and few-shot generalization performance on the classification of grammatical relations for 30 low-resource languages. Languages marked with "∗" are not included in the pretraining corpus. "†" indicates that the language is involved in meta-training for M43. The UD dataset for some languages comprises less than 100 or 200 sentences, and the few-shot performance for these languages is left unspecified.

| | af† | ar | bg | ca | cs‡ | da‡ | de | el | en† | es | et | eu | fa | fi | fr |
|---|---|---|---|---|---|---|---|---|---|---|---|---|---|---|---|
| M28-0 | 78.3 | 85.7 | 90.8 | 89.1 | 87.7 | 86.5 | 89.2 | 88.1 | 84.3 | 89.6 | 83.9 | 82.6 | 88.8 | 85.1 | 91.9 |
| M28-10 | 81.1 | 84.1 | 88.7 | 89.2 | 88.4 | 86.4 | 89.4 | 91.0 | 86.6 | 89.3 | 83.5 | 81.4 | 86.8 | 84.0 | 92.2 |
| M28-30 | 82.4 | 85.1 | 90.8 | 90.1 | 89.2 | 87.3 | 89.9 | 91.8 | 88.1 | 91.0 | 84.3 | 82.3 | 87.6 | 85.3 | 92.7 |
| M28-50 | 83.0 | 86.2 | 91.2 | 90.7 | 89.4 | 88.2 | 90.2 | 92.2 | 87.5 | 91.9 | 84.5 | 83.0 | 89.1 | 85.4 | 93.0 |
| M28-100 | 85.2 | 86.4 | 92.0 | 91.2 | 89.7 | 88.3 | 90.0 | 92.5 | 88.8 | 92.1 | 84.6 | 83.9 | 89.6 | 86.2 | 93.3 |
| M28-200 | 86.0 | 86.6 | 92.3 | 91.9 | 90.1 | 88.6 | 90.2 | 92.9 | 89.7 | 92.5 | 85.2 | 84.3 | 90.2 | 86.5 | 93.4 |
| M43-0 | 85.1 | 84.0 | 89.6 | 88.1 | 87.6 | 86.8 | 88.1 | 92.3 | 86.1 | 88.8 | 81.9 | 80.7 | 81.8 | 83.5 | 91.6 |
| M43-10 | 84.9 | 84.0 | 89.4 | 89.1 | 88.6 | 87.3 | 88.9 | 92.5 | 88.8 | 89.3 | 82.1 | 81.0 | 86.6 | 83.4 | 92.0 |
| M43-30 | 87.0 | 84.2 | 90.8 | 90.0 | 89.1 | 87.3 | 89.9 | 93.4 | 89.5 | 91.7 | 83.0 | 81.9 | 87.5 | 84.3 | 92.5 |
| M43-50 | 87.4 | 85.7 | 90.8 | 90.5 | 89.2 | 87.4 | 90.0 | 93.8 | 90.4 | 91.8 | 83.4 | 82.8 | 88.6 | 85.4 | 92.7 |
| M43-100 | 87.8 | 85.8 | 91.6 | 91.1 | 89.7 | 87.9 | 90.1 | 94.2 | 91.3 | 92.0 | 84.3 | 83.2 | 89.2 | 85.6 | 93.2 |
| M43-200 | 88.3 | 86.3 | 92.1 | 91.6 | 90.1 | 88.1 | 90.2 | 94.4 | 91.5 | 92.3 | 84.7 | 83.8 | 89.8 | 86.0 | 93.3 |

| | ga† | he | hi | hu† | hy† | id† | is | it | ja | ko | lt† | lv | nl | no | pl |
|---|---|---|---|---|---|---|---|---|---|---|---|---|---|---|---|
| M28-0 | 64.6 | 83.0 | 86.7 | 80.5 | 77.9 | 74.0 | 81.9 | 93.0 | 85.4 | 72.7 | 79.1 | 84.0 | 89.4 | 90.5 | 88.6 |
| M28-10 | 67.8 | 81.2 | 84.6 | 83.0 | 78.4 | 77.2 | 77.8 | 92.5 | 84.3 | 66.8 | 80.6 | 83.1 | 89.1 | 89.7 | 87.8 |
| M28-30 | 70.1 | 83.4 | 86.7 | 84.4 | 79.2 | 78.7 | 81.2 | 93.7 | 86.8 | 69.9 | 81.3 | 84.3 | 89.9 | 90.6 | 89.0 |
| M28-50 | 70.8 | 84.6 | 87.1 | 85.4 | 80.1 | 79.7 | 82.3 | 93.8 | 87.4 | 71.5 | 80.8 | 84.4 | 89.8 | 90.7 | 89.3 |
| M28-100 | 71.4 | 85.8 | 87.7 | 86.2 | 80.1 | 81.0 | 83.7 | 94.0 | 89.0 | 74.9 | 82.0 | 85.3 | 90.5 | 91.5 | 90.0 |
| M28-200 | 72.5 | 86.0 | 88.3 | 86.8 | 80.8 | 81.9 | 84.5 | 94.3 | 89.7 | 76.1 | 82.3 | 85.6 | 91.0 | 91.6 | 90.6 |
| M43-0 | 75.8 | 80.0 | 78.9 | 84.1 | 80.0 | 79.1 | 78.0 | 92.7 | 76.3 | 65.4 | 80.9 | 82.8 | 87.6 | 88.8 | 87.9 |
| M43-10 | 75.5 | 81.2 | 84.1 | 86.5 | 80.5 | 81.7 | 77.5 | 93.1 | 81.0 | 64.8 | 81.7 | 82.9 | 88.8 | 88.7 | 87.4 |
| M43-30 | 77.3 | 83.2 | 86.8 | 87.3 | 81.5 | 83.4 | 80.5 | 93.3 | 85.7 | 69.5 | 81.9 | 84.0 | 89.8 | 90.8 | 89.0 |
| M43-50 | 78.0 | 84.0 | 87.4 | 87.4 | 81.6 | 84.5 | 81.3 | 93.6 | 87.3 | 72.4 | 83.4 | 84.2 | 89.9 | 90.9 | 88.6 |
| M43-100 | 78.0 | 85.1 | 87.9 | 88.0 | 82.5 | 85.1 | 83.0 | 94.1 | 87.8 | 74.3 | 84.1 | 85.3 | 90.4 | 91.3 | 89.8 |
| M43-200 | 79.0 | 85.7 | 88.3 | 88.3 | 83.0 | 85.6 | 83.9 | 94.5 | 88.6 | 75.1 | 84.4 | 85.6 | 91.0 | 91.5 | 90.4 |

| | pt | ro | ru | sk | sl‡ | sr† | sv | tr | uk | ur† | vi† | zh |
|---|---|---|---|---|---|---|---|---|---|---|---|---|
| M28-0 | 89.1 | 88.8 | 91.2 | 90.7 | 87.5 | 88.4 | 91.0 | 78.4 | 88.6 | 75.3 | 59.5 | 81.4 |
| M28-10 | 91.4 | 88.3 | 91.5 | 88.9 | 88.1 | 90.0 | 90.6 | 76.5 | 87.9 | 76.0 | 65.9 | 77.8 |
| M28-30 | 93.0 | 89.2 | 92.2 | 91.0 | 89.4 | 90.1 | 91.3 | 77.9 | 88.3 | 77.5 | 68.4 | 81.6 |
| M28-50 | 93.4 | 89.4 | 92.0 | 91.2 | 89.8 | 91.5 | 91.6 | 78.7 | 89.4 | 77.7 | 69.3 | 83.8 |
| M28-100 | 93.7 | 90.4 | 92.1 | 92.4 | 90.5 | 91.8 | 91.9 | 80.0 | 89.9 | 80.0 | 70.7 | 85.2 |
| M28-200 | 93.8 | 90.6 | 92.7 | 92.9 | 90.8 | 92.0 | 92.1 | 80.2 | 90.5 | 80.9 | 72.8 | 86.1 |
| M43-0 | 87.6 | 87.6 | 90.4 | 90.4 | 87.1 | 90.7 | 89.7 | 75.0 | 87.6 | 73.0 | 68.6 | 76.2 |
| M43-10 | 90.4 | 87.9 | 91.2 | 90.6 | 87.6 | 91.5 | 90.1 | 76.6 | 88.2 | 81.4 | 72.9 | 74.2 |
| M43-30 | 92.3 | 89.2 | 92.0 | 91.2 | 89.9 | 92.2 | 90.9 | 77.1 | 89.5 | 82.6 | 75.4 | 81.7 |
| M43-50 | 92.9 | 89.4 | 91.8 | 91.6 | 89.9 | 92.5 | 91.2 | 78.4 | 89.5 | 83.1 | 75.9 | 83.1 |
| M43-100 | 93.4 | 89.9 | 92.5 | 93.0 | 90.0 | 93.1 | 91.8 | 78.9 | 90.2 | 83.4 | 77.4 | 84.5 |
| M43-200 | 93.9 | 90.4 | 92.6 | 93.2 | 90.8 | 93.5 | 92.0 | 79.8 | 90.4 | 84.0 | 78.5 | 85.4 |

Table 9: The zero-shot and few-shot generalization performance on classifying grammatical relations for 42 other languages. By default, these languages are used for meta-training. Languages marked with "†" is involved in meta-training for M43 but not for M28. "‡" indicates that the language is excluded from meta-training for both M28 and M43.

| Language | Abbr. | Language Family | UD Treebanks | Train | Test |
|---|---|---|---|---|---|
| Arabic | ar | Afro-Asiatic.Semitic | PADT | 6,075 | 680 |
| Bulgarian‡ | bg | IE.Balto-Slavik | BTB | 8,907 | 1,116 |
| Catalan‡ | ca | IE.Romance | AnCora | 13,123 | 1,846 |
| German‡ | de | IE.Germanic | GSD | 13,814 | 977 |
| English‡ | en | IE.Germanic | EWT | 12,543 | 2,077 |
| Spanish‡ | es | IE.Romance | GSD | 14,187 | 426 |
| Estonian | et | Uralic.Finnic | EDT | 24,632 | 3,214 |
| Basque | eu | Basque | BDT | 5,396 | 1,799 |
| Persian | fa | IE.Indo-Iranian | PerDT | 26,196 | 1,455 |
| Finnish‡ | fi | Uralic.Finnic | TDT | 12,217 | 1,555 |
| French‡ | fr | IE.Romance | GSD | 14,449 | 416 |
| Hebrew | he | Afro-Asiatic.Semitic | HTB | 5,241 | 491 |
| Hindi‡ | hi | IE.Indo-Iranian | HDTB | 13,304 | 1,684 |
| Icelandic | is | IE.Germanic | Modern | 5,376 | 768 |
| Italian‡ | it | IE.Romance | ISDT | 13,121 | 482 |
| Japanese‡ | ja | Japonic | GSD | 7,050 | 543 |
| Korean | ko | Koreanic | Kaist | 23,010 | 2,287 |
| Latvian | lv | IE.Balto-Slavic | LVTB | 12,521 | 2,325 |
| Dutch‡ | nl | IE.Germanic | Alpino | 12,289 | 596 |
| Norwegian | no | IE.Germanic | Nynorsk | 14,174 | 1,511 |
| Polish‡ | pl | IE.Balto-Slavic | PDB | 17,722 | 2,215 |
| Portuguese‡ | pt | IE.Romance | GSD | 9,615 | 1,200 |
| Romanian‡ | ro | IE.Romance | RRT | 8,043 | 729 |
| Russian‡ | ru | IE.Balto-Slavic | GSD | 3,850 | 601 |
| Slovak | sk | IE.Balto-Slavic | SNK | 8,483 | 1,061 |
| Swedish‡ | sv | IE.Germanic | Talbanken | 4,303 | 1,219 |
| Turkish‡ | tr | Turkic.Oghuz | BOUN | 7,803 | 979 |
| Ukrainian‡ | uk | IE.Balto-Slavik | IU | 5,496 | 892 |
| Chinese (Mandarin)‡ | zh | Sino-Tibetan.Sinitic | GSDSimp | 3,997 | 500 |

Table 10: Languages and UD Treebanks involved in the meta-training of M28, which are typically high-resource languages. English is employed as the source language. Languages marked with "‡" are included in the experiments of in-context learning. The phylogenetic information is obtained from Glottolog (Hammarström et al., 2022). IE stands for the Indo-European family.

| Language | Abbr. | Language Family | UD Treebank | Train | Test |
|---|---|---|---|---|---|
| Afrikaans† | af | IE.Germanic | AfriBooms | 1,315 | 425 |
| Assyrian* | aii | Afro-Asiatic.Semitic | AS | 0 | 57 |
| Akkadian* | akk | Afro-Asiatic.Semitic | PISANDUB | 0 | 101 |
| Amharic* | am | Afro-Asiatic.Semitic | ATT | 0 | 1,074 |
| Belarusian | be | IE.Balto-Slavic | HSE | 22,853 | 1,077 |
| Bhojpuri* | bho | IE.Indo-Iranian | BHTB | 0 | 357 |
| Bambara* | bm | Mande.Western Mande | CRB | 0 | 1,026 |
| Breton* | br | IE.Celtic | KEB | 0 | 888 |
| Buryat* | bxr | Mongolic-Khitan.Mongolic | BDT | 19 | 908 |
| Czech | cs | IE.Balto-Slavic | PDT | 68,495 | 10,148 |
| Welsh† | cy | IE.Celtic | CCG | 976 | 953 |
| Danish‡ | da | IE.Germanic | DDT | 4,383 | 565 |
| Greek† | el | IE.Greek | GDT | 1,662 | 456 |
| Faroese* | fo | IE.Germanic | OFT | 0 | 1,208 |
| Irish† | ga | IE.Celtic | IDT | 4,005 | 454 |
| Swiss German* | gsw | IE.Germanic | UZH | 0 | 100 |
| Mbya Guarani* | gun | Tupian.Maweti-Guarani | Thomas + Dooley | 0 | 98 + 1,046 |
| Croatian‡ | hr | IE.Balto-Slavik | SET | 6,914 | 1,136 |
| Upper Sorbian* | hsb | IE.Balto-Slavik | UFAL | 23 | 623 |
| Hungarian†‡ | hu | Uralic.Hungarian | Szeged | 910 | 449 |
| Armenian† | hy | IE.Armenic | ArmTDP | 1,974 | 277 |
| Indonesian† | id | Austronesian.Malayo-Polynesian | GSD | 4,482 | 557 |
| Kazakh | kk | Turkic.Kipchak | KTB | 31 | 1,047 |
| Kurmanji* | kmr | IE.Indo-Iranian | MG | 20 | 734 |
| Komi-Permyak* | koi | Uralic.Permian | UH | 0 | 100 |
| Komi-Zyrian* | kpv | Uralic.Permian | Lattice | 0 | 663 |
| Karelian* | krl | Uralic.Finnic | KKPP | 0 | 228 |
| Lithuanian† | lt | IE.Balto-Slavik | ALKSNIS | 2,341 | 684 |
| Moksha* | mdf | Uralic.Mordvin | JR | 0 | 342 |
| Marathi† | mr | IE.Balto-Slavik | UFAL | 373 | 47 |
| Erzya* | myv | Uralic.Mordvin | JR | 0 | 1714 |
| Livvi* | olo | Uralic.Finnic | KKPP | 19 | 106 |
| Naija* | pcm | IE.Germanic | NSC | 7,278 | 972 |
| Sanskrit* | sa | IE.Indo-Iranian | UFAL | 0 | 230 |
| Slovenian‡ | sl | IE.Balto-Slavik | SSJ | 10,903 | 1,282 |
| Serbian†‡ | sr | IE.Balto-Slavik | SET | 3,328 | 520 |
| Tamil† | ta | Dravidian.South | TTB | 400 | 120 |
| Telugu† | te | Dravidian.South | MTG | 1,051 | 146 |
| Tagalog | tl | Austronesian.Malayo-Polynesian | TRG | 0 | 128 |
| Urdu† | ur | IE.Indo-Iranian | UDTB | 4,043 | 535 |
| Vietnamese† | vi | Austroasiatic.Vietic | VTB | 1,400 | 800 |
| Warlpiri* | wbp | Pama-Nyungan.Desert Nyungic | UFAL | 0 | 55 |
| Yoruba* | yo | Atlantic-Congo.Volta-Congo | YTB | 0 | 318 |
| Cantonese* | yue | Sino-Tibetan.Sinitic | HK | 0 | 1,004 |

Table 11: The remaining languages and their UD treebanks that are employed in our experiments. Languages marked with "†" are involved in the meta-training of M43. "∗" indicates that the language is not included in the pretraining corpus of mBERT. "‡" shows that the language is included in the experiments of in-context learning. The phylogenetic information is obtained from Glottolog (Hammarström et al., 2022). IE stands for the Indo-European family.