# OpenReview forum: "Are Structural Concepts Universal in Transformer Language Models? Towards Interpretable Cross-Lingual Generalization"
_EMNLP/2023/Conference — EMNLP 2023 Findings_

### Official Review · Reviewer_McsH · 2023-07-30

**Soundness:** 3

**Excitement:**

3: Ambivalent: It has merits (e.g., it reports state-of-the-art results, the idea is nice), but there are key weaknesses (e.g., it describes incremental work), and it can significantly benefit from another round of revision. However, I won't object to accepting it if my co-reviewers champion it.

**Paper Topic And Main Contributions:**

This work investigate how structural concepts are encoded in pre-trained LMs, and whether and how to make them aligned across different languages. POS tags and grammatical relations are used as the concepts for analyzing. The authors first show that the structural concepts are implicitly aligned through vanilla pre-training. Moreover, a meta-learning-based approach is adopted to bring improvements through explicit alignments. Finally, the analysis is extended to the in-context learning setting, where aligning with demonstration examples brings further gains.

**Questions For The Authors:**

- A: I’m not sure whether LLaMA is a good model for multi-lingual analysis (most pre-training data is in English), maybe a multi-lingual LLM such as BLOOM might be more suitable?
- B: It might be interesting to analyze the correlations between alignability and language distances, has this been considered in the experiments?


**Reasons To Accept:**

- This work provides a detailed and comprehensive analysis on the cross-lingual alignments of structural concepts in pre-trained LMs and might inspire future work on better understanding of LMs.
- The paper is well-structured and easy to follow.


**Reasons To Reject:**

- The analysis is in some way limited; only two types of syntactic structures are included. It will be more interesting to see more analyses on semantic concepts as well as how the explicit alignments can help further downstream tasks.
- Only two pre-trained LMs (mBERT and LLaMA) are analyzed; some more models should be included to demonstrate that the findings generalize well to different models.


**Reproducibility:**

4: Could mostly reproduce the results, but there may be some variation because of sample variance or minor variations in their interpretation of the protocol or method.

**Reviewer Confidence:**

3: Pretty sure, but there's a chance I missed something. Although I have a good feel for this area in general, I did not carefully check the paper's details, e.g., the math, experimental design, or novelty.

---

> ### Author Rebuttal · Authors · 2023-08-28
>
> Thank you for your comments. Before answering the questions, we would like to elucidate some central points of our paper, which may improve the understanding of our following response.
>
> - This paper aims to improve cross-lingual generalization to truly low-resource languages that are distant from high-resource ones. This presents a big challenge because i) there lacks sufficient data to support learning and ii) the substantial cross-linguistic differences render it difficult to apply the knowledge acquired from high-resource languages to these low-resource ones.
>
> - To this end, rather than leveraging the cross-linguistic similarities captured by LLMs, we propose to explicitly establish alignment between concepts of which there is **correspondence** across diverse languages. Take spacial concepts like "left," "right," "top," and "bottom" as an example. While these concepts are represented differently across languages, it is possible that the structure of relations between these concepts within different languages is alignable, providing knowledge that can be transferred among languages.
>
> - To investigate its feasibility, we present two approaches: the first is a method to measure the degree to which the correspondence between concepts in different languages is captured by LLMs (Section 2), and the second is a meta-learning-based approach that learns to align the conceptual correspondence between languages in a few-shot (/zero-shot) manner, facilitating cross-lingual generalization particularly towards low-resource languages (Section 3). We use the structural concepts defined in Universal Dependencies as a testbed for our analyses.
>
> Below are our answers to the questions.
>
> **Question#1**: While the majority of LLaMA's pretraining data is in English, it involves Wikipedia dumps spanning 20 languages and exhibits capability of cross-lingual transfer (Section 4.3). Our analysis of its representations suggests that the conceptual correspondence between languages is captured by it (Section 2.3). Moreover, we also extend our analysis to languages not involved in its pretraining data for comparison (Section 2.3 \& Section 4.3). The impact of distribution of languages in pretraining data is an interesting topic and we leave exploration of this for future work.
>
> **Question#2**: We have discussed factors potentially influencing the alignability of structural concepts in Section 2.4 Line 260-279. Our emphasis in this paper lies in the investigation of conceptual correspondence rather than similarities and differences between languages. The subtle differences between languages that impact the alignability between languages is an intriguing topic that warrants deeper future exploration.
>
> Additionally, we would like to clarify some issues raised in **Reasons To Reject**.
>
> Our approaches rely solely on representations derived from LLMs, which makes them model-agnostic and readily adaptable to various models and concepts.
>
> - Regarding the first point, we use the structural concepts outlined in UD including word classes and grammatical relations as a testbed. Through this, we demonstrate the viability to improve generalization to truly low-resource languages by explicitly aligning conceptual correspondence between languages. Our results support the efficacy of our methods. It is intriguing to generalize our method to other contexts like different modalities and semantic facets of language, as discussed in Section 6 (Line 637-647). However, considering space limitations, exploring these broader contexts falls outside the scope of this paper.
>
> - Regarding the second point, our selection includes two extensively used Transformer-based models that are pretrained on multiple languages without explicit cross-lingual information. These two models are trained via distinct yet widely adopted pretraining objectives: masked language modeling and language modeling. The results of our experiments with the models support the efficacy of our approaches. We leave the exploration of alternative models for future work.
>
> We are open to discussing our paper further if there are any remaining questions or concerns.

---

### Official Review · Reviewer_TY8m · 2023-08-04

**Soundness:** 3

**Excitement:**

4: Strong: This paper deepens the understanding of some phenomenon or lowers the barriers to an existing research direction.

**Paper Topic And Main Contributions:**

This paper explores the possibility to improve cross-lingual generalization by aligning structural concepts like word classes and grammatical relations across different languages. By setting a linear transformation (probe) on top of the multiLMs like mBERT and LLaMA, they show that there's alignment between structural concepts across different language.
Given the observed (weak) alignment, the authors try to strengthen the alignment by proposing a meta-learning based approach. The proposed meta-learning method seems data-efficient and is competitive to the SoTA (and even outperforms it for certain low-resource languages).
Finally they analyse LLaMA performance for POS tagging in the in-context learning (ICL) setting, where they argue that although in-domain performance is quite high, the generalization to other languages is not high. Therefore they use the proposed meta-learning approach once more but in the ICL setting to help the model improve alignment of structural concepts across languages. Their method achieves considerable gains in generalization compared to the baseline ICL when the proposed method is applied.

**Questions For The Authors:**

- how did you tune the values for h and m for the language-agnostic function? Given figure 2, it seems that even much lower values for `m` could work?
- Given the proposed approach, how can we ensure if e.g. RSA is high due to model's internal multilingual alignment, or due to having the transformation matrix A which could strengthen the alignment?

**Reasons To Accept:**

- The paper is well-written and easy to follow
- The observation of syntactic alignment between structural concepts for multiLMs in different scales seem promising. There also has been a thorough study by the authors with two metrics over two models at different scales to demonstrate the (relative) alignment.
- The proposed method can get competitive results to SoTA without updating the multiLM parameters, and especially has promising performance for low-resource languages. It's also shown that the proposed meta-learning approach is quite data efficient (compared to FT), and few samples in the target low-resource languages considerably improves generalization.

**Reasons To Reject:**

- Some of the results in the paper doesn't seem to be a fair comparison. For instance, in Figure 5, Meta-training is seemingly performed on many languages, while `En` and `Mono` using only a single language in training data. Similar concern for Figure 3 as well. It's also evident than when a model like UDapter is studied in Table 1 (which has a more fair training data), the performance is even a bit better than the proposed meta-learning approach.
- Procrustes and RSA values in the figure 1 seems "contradictory" for the mBERT model. There's no explanation in the paper why this inconsistency occurs

**Reproducibility:**

4: Could mostly reproduce the results, but there may be some variation because of sample variance or minor variations in their interpretation of the protocol or method.

**Reviewer Confidence:**

2: Willing to defend my evaluation, but it is fairly likely that I missed some details, didn't understand some central points, or can't be sure about the novelty of the work.

---

> ### Author Rebuttal · Authors · 2023-08-28
>
> Thank you for your detailed comments. Before answering the questions, we would like to elucidate some central points of our paper, which may improve the understanding of our following response.
>
> - In this paper, we propose to explicitly establish alignment between concepts of which there is **correspondence** across diverse languages, thereby facilitating cross-lingual generalization, particularly towards low-resource languages. Take spacial concepts like "left," "right," "top," and "bottom" as an example. While these concepts are represented differently across languages, it is possible that the structures of relations between these concepts within different languages is alignable, providing knowledge that can be transferred among languages. We use the structural concepts defined in Universal Dependencies as a testbed for our analyses.
>
> - We show that the structural concepts represented by LLMs can be defined based on their prototypes. We further demonstrate that the relational structures of the concepts within different languages are highly alignable, which indicates that the correspondence between the concepts is captured by LLMs (Section 2).
>
> - As the correspondence between concepts holds across languages, we propose a meta-learning-based approach that learns to align the conceptual correspondence in a few-shot (/zero-shot) manner, facilitating generalization particularly to low-resource languages (Section 3).
>
> Below are our answers to the questions.
>
> **Question#1**: The value of $m$ is determined based on the findings in Section 2.3. Specifically, $m = 32$ for word class identification and $64$ for grammatical relation, where the probe accuracy approaching its peak performance. $h$ is selected to be larger than $m$. We examine cases where $h \in \left\\{128, 256, 512 \right\\}$ and observe little improvements by increasing $h$ beyond $256$.
>
> **Question#2**: Rather than assessing the LLMs' internal alignment, we use RSA to measure whether correspondence between structural concepts is encoded in LLMs. The matrix $A$ is a simple linear transformation exclusively trained to derive features of structural concepts from LLMs, which enables classification of samples according to their distances to prototypes for each concept. RSA is employed to evaluate the degree to which the relational structures of the prototypes within different languages are alignable.
>
> In addition, we would like to clarify some issues raised in **Reasons To Reject**.
>
> Regarding the first point:
>
> - **Figure 5 and Figure 3** are meant to illustrate the performance gain achieved by our proposed meta-learning-based method. This shows that the structural concepts are effectively aligned through our method, and it effectively generalizes to languages unseen during meta-training, as mentioned in Section 4.3 Line 531-534 and Section 3.3 Line 381-386.
>
> - Based on typological information, **Udapter** effectively exploits cross-linguistic similarities to implicitly transfer knowledge between languages. Through adapter-based tuning, (which has shown to achieve comparable performance to fine-tuning methods,) it achieves SOTA performance on high-resource languages like English, as well as languages similar to high-resource ones. By contrast, we focus on the generalization to low-resource languages that are distant from high-resource ones. By explicitly leveraging conceptual correspondence between languages, our approach offers particular advantages in terms of these languages, while solely relying on representations derived from pretrained LLMs. The results support the efficacy of our approach (Line 363-371).
>
> We appreciate your comments and will revise our manuscript concerning the figures and the table mentioned here to enhance clarity.
>
> Regarding the second point:
>
> - RSA and Procrustes analysis are two complementary methods with distinct metrics. Consequently, complete uniformity between the results is not anticipated. RSA is a non-parametric method to compare the geometry of two representation spaces. In our context, for two languages, RSA computes the rank correlation between the relational structure of the concepts within each language. By contrast, Procrustes analysis evaluates the potential for **linear** alignment between two spaces by finding the optimal orthogonal transformation that matches corresponding representations (of concepts) in the two spaces. It is possible that the relational structure within two spaces are similar but they cannot be aligned linearly. Experiment results derived from both methods suggest that the correspondence between structural concepts across languages is encoded in LLMs.
>
> We are open to discussing our paper further if there are any remaining questions or concerns.

---

### Official Review · Reviewer_m4pp · 2023-08-04

**Soundness:** 4

**Excitement:**

3: Ambivalent: It has merits (e.g., it reports state-of-the-art results, the idea is nice), but there are key weaknesses (e.g., it describes incremental work), and it can significantly benefit from another round of revision. However, I won't object to accepting it if my co-reviewers champion it.

**Paper Topic And Main Contributions:**

In this paper, they explicitly align conceptual correspondence (syntactic aspect) across languages to enhance cross-lingual transfer. They conduct experiments on 43 languages, where they show that: 1) a high degree of alignability for syntactic concepts can be deduced from LMs, 2) a proposed meta-learning-based method, targeting align concepts explicitly, can enhance cross-lingual transfer for downstream syntactic tasks, like word class and grammatical relations identification. 3) for in-context learning, the transfer can be achieved as well. The experiments are sufficient for their claim, although there are a little few places I think can be made clearer.

**Questions For The Authors:**

1. How about the performance of static word embedding for concept alignment as you did in section-2? See point 2 in Reason to Reject.
2. Line 304-308: why do you think a language-specific linear mapping can achieve a similar effect that orthogonal transformation gets?
3. Can you briefly describe the meta-learning setting in sec-3.1, e.g., what is the objective? Which meta-learning approach is used? During the meta-testing time, whether cross-lingual label signals are leveraged? (I guess so according to lines 327-329).

I will be happy to raise my score if questions get answered convincingly.

**Reasons To Accept:**

1. Good writing skills, although a little bit unclear in some places.
2. Sufficient experimental demonstration. A good effect on cross-lingual transfer is shown.

**Reasons To Reject:**

1. The setting of meta-learning in sec-3.1 is not described clearly.

2. Part of the baselines is not convincing to me: For the experiments in 2.2, they conduct 3 baselines by randomly choosing other concepts or samples. Can we look at concepts deduced from static word embeddings, which do not have any structural information like word2vec or fasttext. Then, let’s say how better alignability of structural concepts deduced from LLMs (contextual embedding) can be achieved compared to it. It may be impossible for classification but for alignments it might be fine. I am really curious about this, which is related to the main contribution of this paper as I see. If it is questionable, hard to say where the transfer exactly comes from.

**Reproducibility:**

4: Could mostly reproduce the results, but there may be some variation because of sample variance or minor variations in their interpretation of the protocol or method.

**Reviewer Confidence:**

4: Quite sure. I tried to check the important points carefully. It's unlikely, though conceivable, that I missed something that should affect my ratings.

---

> ### Author Rebuttal · Authors · 2023-08-28
>
> Thank you for your detailed comments.
>
> **Question#1**: This point is intriguing and merits consideration. However, previous work (e.g., \[1\]) has shown that structural information like grammatical relations and word classes (POS) can be decoded from static word embeddings (produced by, e.g., skip-gram and C\&W models) to some extent. Our results in Appendix A.1 Fig. 8 \& 9 also show that structural information can be partly derived from the 0-th layer of mBERT even in the absence of contextual information, consistent with \[2\].
>
> In addition, we would like to clarify some issues raised in **the second point in Reasons To Reject**.
>
> - The baselines in Section 2 are constructed to validate the **significance of alignability** between structural concepts in different languages, which is akin to previous work in validating the significance of the results measured by methods like RSA and Procrustes analysis (e.g., \[3, 4, 5\]).
>
> - It is the correspondence between structural concepts in different languages that we aim to leverage for cross-lingual transfer. Given a source-target language pair and sufficient labeled data, we may directly learn a linear (orthogonal) transformation to align the conceptual correspondence, allowing classification of samples in the target language based on the prototypes in the source language. As the conceptual correspondence holds across languages, we rely on meta-learning to learn to align the conceptual correspondence in a few-shot manner, thereby generalizing to target languages with minimal labeled data. Our approach enables generalization to languages unseen during meta-training, particularly benefiting low-resource languages that are distant from high-resource ones.
>
> **Question#2**: Given the results from Procrustes analysis in Section 2, the prototypes in different languages can be aligned through an orthogonal transformation, which is indeed a linear transformation that preserves the lengths of vectors and angles between them. Through an orthogonal transformation that aligns the prototypes in a source and a target language, the samples in the target language can be classified according to their distances to the prototypes in the source language. We use a linear mapping to approximate the orthogonal transformation, akin to previous work that learns to align word embeddings in different languages \[6, 7, 8, among others\].
>
> **Question#3**: In Section 3.1, we propose a meta-learning-based method to learn to align the conceptual correspondence between languages, which effectively facilitates generalization to languages not encountered during meta-training. We first derive the prototypes in the source language through **labeled** sentences in its training dataset. During meta-training, a training episode consists of a support set with $N$ **labeled** sentences and a query set with $M$ **labeled** sentences in a certain (target) language. The details can be found in Appendix D Line 1294-1307. The language-agnostic function $f_{\phi}$ is optimized over the entire training procedure, and the language-specific function $g_{\alpha}$ is learned separately for different languages. During meta-testing, $f_{\phi}$ is kept fixed and $g_{\alpha}$ is trained from scratch using the available examples (i.e., a support set with $N$ **labeled** sentences in $L_{T}$) in the few-shot setting. The method here bears resemblance to \[9\]. We appreciate your comments and will incorporate the descriptions of the experimental setting into our manuscript to improve clarity.
>
> We are open to discussing our paper further if there are any remaining questions or concerns.
>
> \[1\] Qian, P., Qiu, X., & Huang, X. (2016). Investigating Language Universal and Specific Properties in Word Embeddings. ACL.
>
> \[2\] Chi, E. A., Hewitt, J., & Manning, C. D. (2020). Finding Universal Grammatical Relations in Multilingual BERT. ACL.
>
> \[3\] Kriegeskorte, N., Mur, M., & Bandettini, P. A. (2008). Representational similarity analysis-connecting the branches of systems neuroscience. Frontiers in systems neuroscience, 4.
>
> \[4\] Patel, R., & Pavlick, E. (2021). Mapping language models to grounded conceptual spaces. ICLR.
>
> \[5\] Abdou, M., Kulmizev, A., Hershcovich, D., Frank, S., Pavlick, E., & Søgaard, A. (2021). Can Language Models Encode Perceptual Structure Without Grounding? A Case Study in Color. CoNLL.
>
> \[6\] Mikolov, T., Le, Q. V., & Sutskever, I. (2013). Exploiting similarities among languages for machine translation. arXiv:1309.4168.
>
> \[7\] Lample, G., Conneau, A., Ranzato, M. A., Denoyer, L., & Jégou, H. (2018). Word translation without parallel data. ICLR.
>
> \[8\] Schuster, T., Ram, O., Barzilay, R., & Globerson, A. (2019). Cross-Lingual Alignment of Contextual Word Embeddings, with Applications to Zero-shot Dependency Parsing. NAACL-HLT.
>
> \[9\] Li, W. H., Liu, X., & Bilen, H. (2022). Cross-domain few-shot learning with task-specific adapters. CVPR.

---

### Meta-Review · Area_Chair_iLw3 · 2023-09-19

**Recommendation:** 4

**Metareview:**

There is consensus amongst the reviewers that the work is sound and well written. The majority of the reviewers agree that the experimental design and analysis is sufficiently comprehensive, and that all the claims made in the paper are well-supported. The general excitement is moderate, with the text of the reviews (and reasons to accept) not particular championing for the idea and its novelty itself. Overall, the positives of the paper outweigh the weaknesses which were not addressed in the rebuttal, and the paper has _good technical soundness_ and _ambivalent excitement_.

The following is a summary of the strengths, weaknesses and scores across the three reviews:

**Strengths:**

- Paper is well written (**m4pp**, **TY8m**, **McsH**)
- Experimental design is sufficient in backing the claims made in the paper (**m4pp**)
- Adaptation of proposed technique (meta-learning approach) to in-context learning is especially pertinent, and shows the efficacy of the proposed methodology in low-resource settings (**TY8m**)
- Analysis of cross-lingual alignment is comprehensive (**McsH**, **m4pp**)

**Weaknesses:**

- Baselines are limited and too simple (**m4pp**)
	- Rebuttal somewhat addresses this weakness
- Generalization of the findings in general is not sufficiently shown (**McsH**)
	- e.g. two models, one of which was not explicitly trained on non-english

**Scores in decreasing order of confidence:**

|      | Soundness | Excitement | Reproducibility | Confidence |
|------|-----------|------------|-----------------|------------|
| m4pp | 4         | 3          | 4               | 4          |
| McsH | 3         | 3          | 4               | 3          |
| TY8m | 3         | 4          | 4               | 2          |

---

### Decision · Program_Chairs · 2023-10-07

**Decision:**

Accept-Findings

**Comment:**

There is consensus amongst the reviewers that the work is sound and well written. The majority of the reviewers agree that the experimental design and analysis is sufficiently comprehensive, and that all the claims made in the paper are well-supported. The general excitement is moderate, with the text of the reviews (and reasons to accept) not particular championing for the idea and its novelty itself. Overall, the positives of the paper outweigh the weaknesses which were not addressed in the rebuttal, and the paper has _good technical soundness_ and _ambivalent excitement_.

The following is a summary of the strengths, weaknesses and scores across the three reviews:

**Strengths:**

- Paper is well written (**m4pp**, **TY8m**, **McsH**)
- Experimental design is sufficient in backing the claims made in the paper (**m4pp**)
- Adaptation of proposed technique (meta-learning approach) to in-context learning is especially pertinent, and shows the efficacy of the proposed methodology in low-resource settings (**TY8m**)
- Analysis of cross-lingual alignment is comprehensive (**McsH**, **m4pp**)

**Weaknesses:**

- Baselines are limited and too simple (**m4pp**)
	- Rebuttal somewhat addresses this weakness
- Generalization of the findings in general is not sufficiently shown (**McsH**)
	- e.g. two models, one of which was not explicitly trained on non-english

**Scores in decreasing order of confidence:**

|      | Soundness | Excitement | Reproducibility | Confidence |
|------|-----------|------------|-----------------|------------|
| m4pp | 4         | 3          | 4               | 4          |
| McsH | 3         | 3          | 4               | 3          |
| TY8m | 3         | 4          | 4               | 2          |